# A Comparison of Plume Rise Algorithms to Stack Plume Measurements in the Athabasca Oil Sands

Mark Gordon[1], Paul A. Makar[2], Ralf M. Staebler[3], Junhua Zhang[2], Ayodeji Akingunola[2], Wanmin Gong[2], Shao-Meng Li[3]

1) Earth and Space Science and Engineering, York University
2) Air Quality Modelling and Integration Section, Air Quality Research Division, Atmospheric Science and Technology Directorate, Science and Technology Branch, Environment and Climate Change Canada.
3) Air Quality Processes Research Section, Air Quality Research Division, Atmospheric Science and Technology Directorate, Science and Technology Branch, Environment and Climate Change Canada.

## Abstract

Plume rise parameterizations calculate the rise of pollutant plumes due to effluent buoyancy and exit momentum. Some form of these parameterizations is used by most air quality models. In this paper, the performance of the commonly used Briggs plume rise algorithm was extensively evaluated, through a comparison of the algorithm's results when driven by meteorological observations with direct observations of plume heights in the Athabasca oil sands region. The observations were carried out as part of the Canada-Alberta Joint Oil Sands Monitoring Plan in August and September of 2013. Wind and temperature data used to drive the algorithm were measured in the region of emissions from various platforms, including two meteorological towers, a radio-acoustic profiler, and a research aircraft. Other meteorological variables used to drive the algorithm include friction velocity, boundary-layer height, and the Obukhov length. Stack emissions and flow parameter information reported by Continuous Emissions Monitoring Systems (CEMS) were used to drive the plume rise algorithm. The calculated plume heights were then compared to interpolated aircraft $SO_2$ measurements, in order to evaluate the algorithm's prediction for plume rise. We demonstrate that the Briggs algorithm, when driven by ambient observations, significantly underestimated plume rise for these sources, with more than 50% of the predicted plume heights falling below half the observed values from this analysis. With the inclusion of the effects of effluent momentum, the choice of different forms of parameterizations, and the use of different stability classification systems, this essential finding remains unchanged. In all cases, approximately 50% or more of the predicted plume heights fall below half the observed values. These results are in contrast to numerous plume rise measurement studies published between 1968 and 1993. We note that the observations used to drive the algorithms imply the potential presence of significant spatial heterogeneity in meteorological conditions; we examine the potential impact of this heterogeneity in our companion paper (Akingunola et al, 2018). It is suggested that further study using long-term in-situ measurements with currently available technologies is warranted to investigate this discrepancy, and that wherever possible, driving meteorological observations are conducted in the immediate vicinity of the emitting stacks.

## 1. Introduction

In large scale air-quality models, grid cell sizes may be on the order of 1 km or larger, while vertical resolution may be in the tens to hundreds of meters (c.f. Makar *et al.*, 2015a,b). The large-scale impacts of transport by winds and turbulence are handled in these models by algorithms dealing with advection and turbulent diffusion of tracers. However, the redistribution of mass from elevated stacks with high-temperature and/or high-velocity emissions sources requires parameterization in order to deal with issues such as the buoyancy and momentum of the emitted mass. Briggs and others developed a system of parameterizations for plume rise beginning in the late 1960's (e.g. Briggs, 1969; Briggs, 1975). The parameterizations followed dimensional analysis to estimate plume rise based on meteorological measurements, atmospheric conditions, and stack parameters. Different variations of the Briggs plume rise parameterization equations are used in three-dimensional air-quality models such as GEM-MACH (Makar *et al.*, 2015a,b), CMAQ (Byun and Ching, 1999), CAMx (Emery *et al.*, 2010), as well as AEROPOL, SCREEN3, and CALGRID models (see Holmes and Morawska, 2006 for a summary of these models). The Briggs equations are also used in the Regional Acid Deposition Model (RADM, Byun and Binowski, 1991), and have been incorporated into emissions processing systems such as SMOKE (CMAS Website) and SMOKE-EU (Bieser *et al.*, 2011a).

As summarized by Briggs (1969), early observation of plume rise incorporated a wide variety of methods. Plumes were visually traced on Plexiglas screens, photographed, compared in height to nearby towers, and measured with lidar. Other techniques included the release of Geiger counters attached to balloons, and the release of balloons from within the stack chimneys. Bringfelt (1968) summarizes other techniques, using either theodolite, cloud height searchlights, or fluorescent particles sampled by aircraft-mounted instruments. Scaled wind tunnel simulations were also used. These observations were used to constrain the plume rise parameterizations and to choose appropriate constants following dimensional analysis (see Bieser *et al.*, 2011b for a summary).

Once a set of equations for plume rise had been developed, further observations were used to test their accuracy. A report of these comparisons (VDI, 1985) summarizes five studies in which plume rise parameterizations were compared to observations. These studies consistently show a tendency to overestimate plume rise using the Briggs parameterizations. Giebel (1979) measured pit coal power plant plumes with lidar which averaged 50% lower than the parameterization. Rittmann (1982) reanalyzed the Bringfelt (1968) and Briggs (1969) measurements from "industrial-sized sources" and found most plume heights were between 12 and 50% of the predicted rise. England *et al.* (1976) measured plume rise at a gas turbine facility with airborne measurements of $NO_x$ and found plumes were 30% lower than predicted. Hamilton (1967) measured power station plumes with lidar which averaged 50% lower than the parameterization. Moore (1974) used data from seven locations measured with a variety of methods (photography, lidar, aircraft, and balloons) and found measured plume rise was 10-20% lower than the parameterization. The authors of the VDI (1985) report recommend reducing the plume height predicted by the Briggs equations by 30% during neutral conditions. No recommended adjustment for stable and unstable conditions was proposed, primarily due to a lack of supporting data. Sharf *et al.* (1993) measured the rise of power plant plumes with

aircraft-based $SO_2$ measurements and found that plume heights were generally overestimated by the parameterization by up to 400 m.  More recently, Webster and Thompson (2002) tested the Briggs equations as well as a more complex Lagrangian model using a network of surface concentration measurements downwind of a power plant.  The Briggs algorithm resulted in concentration predictions which were biased high relative to observations, potentially indicating a tendency to underestimate plume rise, as emissions distributed over a lower vertical height would result in higher concentration.  However, there may be other factors leading to the overestimation, such as poorly modelled winds or overestimated emission rates.  Hence, the majority of earlier studies which have been compared to the original Briggs plume rise parameterization indicated some degree of overestimation of the actual plume rise, with a single more recent study possibly suggesting an underestimation of actual plume rise (inferred through surface measurements).

In the summer of 2013, as part of the Canada-Alberta Joint Oil Sands Monitoring (JOSM) Plan, aircraft measurements and monitoring stations were used to study dispersion and chemical processing of pollutants emitted from sources in the Athabasca oil sands region of northern Alberta.  The GEM-MACH model (nested to 2.5 km resolution) was run from August through September, coincident with the measurement campaign, as an aid in directing aircraft flights and in subsequent post-campaign analysis of the observations.  The model makes use of the Briggs plume rise algorithms.  The large stacks in the region emit many key pollutants, such as $SO_2$, $NO_x$, VOCs, CO, and aerosols.  The accuracy of the plume calculations thus has significant impact on model predictions, particularly close to the sources.

This manuscript evaluates the performance of the Briggs plume rise parameterization, as it is formulated in Environment and Climate Change Canada's GEM-MACH model, in a "stand-alone/off-line" sense, using meteorological observations as well as stack parameter data to drive the Briggs algorithms.  For comparison, another model proposed by Briggs (1984) for irregular stability profiles is also evaluated.  We also make use of aircraft observations of emitted $SO_2$ in order to evaluate the accuracy of the algorithms.

In our companion paper (Akingunola *et al.* 2018, this issue) we examine the potential impact of the observed heterogeneity in meteorological data on plume rise predictions, comparing high resolution GEM-MACH plume locations to aircraft observations, as well as the effects of different sources of stack data on simulated plume rise performance.

## 2. Methods

### 2.1 Plume Rise Parameterization in GEM-MACH.

The plume rise ($\Delta h$) calculation in GEM-MACH is driven by 9 variables: stack height ($h_s$), exit temperature at the stack outlet ($T_s$), stack emission volumetric flow rate ($V$), air temperature at stack height ($T_a$), wind speed at stack height ($U$), surface temperature ($T_{surface}$), boundary-layer height ($H$), friction velocity ($u_*$), and Obukhov length ($L$). These input parameters are used to generate the rise in the plume above the stack height ($\Delta h$), as well as the upper and lower boundaries of the plume having risen to equilibrium. In models such as GEM-MACH, buoyant transport of emissions through that region is assumed to be instantaneous. The emitted mass is distributed through the given region under the assumption that the buoyant plume has reached equilibrium. Here, all of these variables are obtained from observations (either directly or via the use of the appropriate formulae with observed quantities).

The algorithm makes use of derived quantities (the buoyancy flux, $F_b$, the stability parameter, $s$, and the convective velocity, $H_*$) with different formula for plume rise corresponding to *neutral*, *stable,* and *unstable* atmospheric conditions. The buoyancy flux is calculated from Briggs (1984, equivalent to their Eq. 8.35) as

$$F_b = \begin{cases} \dfrac{g}{\pi} V \dfrac{(T_s - T_a)}{T_s}, & T_s > T_a \\ 0, & T_s \leq T_a \end{cases}, \tag{1}$$

where $g = 9.81$ m s$^{-2}$ is the gravitational acceleration. The stability parameter is calculated from Briggs (1984, combining their Eq. 8.8 and Eq. 8.14) as

$$S = \frac{g}{T_a}\left(\frac{dT_a}{dz} + \frac{g}{c_p}\right). \tag{2}$$

where $z$ is the height coordinate and $c_p = 1005$ J K$^{-1}$ kg$^{-1}$. The temperature gradient is calculated from the temperature difference over the stack-height ($dT/dz = (T_a - T_{surface})/h_s$), with a minimum value set at –5 K/km (i.e. $S \geq 0.047/T_a$). We note that calculating the temperature difference between the stack height and the surface may underestimate the temperature gradient above the stack height, where the plume rises. The extent of this effect is tested later using temperature gradients throughout the boundary layer (Section 2.2). Finally, the convective velocity ($H_* = -2.5u_*^3/L$) is defined in Briggs (1985).

The atmosphere is considered *neutral* if $L > 2h_s$ or $L < -0.25h_s$ (i.e. $-4 < \frac{h_s}{L} < 0.5$ ). These values are suggested in Briggs (1984) and the sensitivity of the results to these values is tested in Section 4. The plume rise in neutral conditions is taken as the minimum of two formulations of Briggs outlined in Sharf *et al.* (1993) and Byun and Ching (1999) as

$$\Delta h = \min\left[39\frac{F_b^{3/5}}{U} \,, \; 1.2\left(\frac{F_b}{u_*^2 U}\right)^{3/5}\left(h_s + 1.3\frac{F_b}{u_*^2 U}\right)^{2/5}\right]. \tag{3}$$

The atmosphere is considered *stable* at the plume height if either $0 < L < 2h_s$ (stable conditions) or $h_s \geq H$ (direct emission above the boundary-layer). From Briggs (1984, their Eq. 8.71), the plume rise is calculated as

$$\Delta h = 2.6 \left(\frac{F_b}{SU}\right)^{\frac{1}{3}}.$$ (4)

The atmosphere is considered *unstable* if $-0.25h_s < L < 0$. In the unstable case, the plume rise is taken as the minimum value of two formulations of Briggs outlined in Byun and Ching (1999),

$$\Delta h = \min\left[3\left(\frac{F_b}{U}\right)^{\frac{3}{5}} H_*^{-\frac{2}{5}}, \quad 30\left(\frac{F_b}{U}\right)^{\frac{3}{5}}\right].$$ (5)

This effectively places a lower limit on the magnitude of the convective velocity in determining plume rise as $H_* > 0.00316$ m$^2$/s$^3$ (from $H_*^{-2/5} < 10$). Briggs (1984) gives the example of clear summer conditions as $H_* = 0.007$ m$^2$/s$^3$.

The only difference between Eqns. 3, 4, and 5 and the plume rise parameterizations used in SMOKE (described in Bieser *et al.*, 2011 and Houyoux, 1998) is the option of the minimum values in unstable and neutral conditions. In the SMOKE model, only the second parameterizations within the minima of Eqns. 3 and 5 are used. Both the approaches used in GEM-MACH and SMOKE are investigated in the following analysis.

Plume rise is also modified for situations where the stack height is less than the boundary-layer height ($h_s < H$), but the plume rises high enough to penetrate the boundary-layer height to some degree ($h_s + \Delta h > H$). This is referred to as "bumping" (Briggs, 1984). The vertical plume depth is assumed to be equal to the plume rise so that the plume is bound by the height range $h_s + 0.5\Delta h < z < h_s + 1.5\Delta h$. If any portion of the plume is above $H$, the plume rise is calculated (from Briggs, 1984) as

$$\Delta h = (0.62 + 0.38p)(H - h_s),$$ (6)

where $p$ is the fraction of the plume above $H$ (i.e. $p = 0$ if $h_s + 1.5\Delta h = H$ and $p = 1$ if $h_s + 0.5\Delta h = H$).

While the above formulae are used in GEM-MACH and other models, we also examine a layer-based approach suggested by Briggs, described below, and the companion paper, Akingunola *et al.* (2018), examines the impact of this approach within the GEM-MACH model itself.

## 2.2 Plume Rise into Irregular Stability Profiles (The Layered Method)

In addition to the parameterization discussed above, Briggs (1984) suggests a layer-based approach to calculate plume rise for complex stability profiles. In this approach, the plume buoyancy ($F$) is modified as it passes through each discrete layer as

$$F = F_j - 0.053 S_j U_j \left(z_c^3 - z_j^3\right)$$ (7)

where $F_j$ is the buoyancy flux at the bottom of layer $j$, $S_j$ is the layer stability calculated using Eq. (2), $U_j$ is the wind speed, and $z_j$ is the layer height above the stack height. The wind speed

in the original Briggs formulation is taken as constant with height, while here we use an average wind speed for each layer. The lower boundary of the first layer is the stack height ($z_{j=0} = 0$). The value of $F$ is determined sequentially for each layer at the top of each layer (with $z_c = z_{j+1}$) until it becomes negative. For the layer where $F$ becomes negative, Eq. 7 is solved to give the plume height $z_c$ for which $F = 0$. Plume rise is calculated as $\Delta h = z_c$. Layer thickness will depend on the vertical model or measurement resolution. Layer thickness for this analysis is discussed in detail in Section 2.6.

Equation 7 is intended for use with stable ($S > 0$) or neutral ($S = 0$) layers. For unstable layers we follow the approach outlined in our companion paper (Akingunola et al., 2018) in which the plume rises through the unstable layer without gaining or losing buoyancy or momentum (equivalent to $S = 0$ in Eq. 7). As is discussed below (Section 4.1), the majority of layer temperature profiles (>90%) measured by the aircraft were stable or neutral, so this assumption should not have a significant effect on the resulting plume rise. However, we also hound that the stability was spatially heterogeneous in the study region, with significant differences in stability noted from the different sources of meteorological information.

While the Briggs parameterization discussed in Section 2.1 is driven by surface (or near-surface) observations, the layered method (Eq. 7) is driven by observations up to the height of the plume. The observed plume centreline heights (Section 2.7) vary between approximately 100 m and 1000 m above the surface. Hence the layered method can be used with the elevated observations from an aircraft measurement platform and an acoustic profiler (Section 2.4).

**2.3 Stack Height ($h_s$), Exhaust Temperature ($T_s$), and Flow Rate ($V$)**

As part of the Continuous Emission Monitoring System (see CEMS, 1998), measurements of 19 stacks in the region of study with valid hourly measurements of $SO_2$ and average effluent velocity and temperature were obtained from Alberta Environment and Parks. Stacks which emit primarily $NO_x$ and no reported $SO_2$ are not used in this analysis. A key requirement for our evaluation is that the stacks selected for comparison have sufficient levels of $SO_2$ emissions to be easily discernable from the aircraft observations. For stacks without reported CEMS $SO_2$ emission rates, the average rates determined from the Cumulative Environmental Management Association inventory for the year 2010 (see CEMA, 2012) were used to eliminate stacks from the analysis which would not emit enough $SO_2$ to be observed by the aircraft-based instrumentation. It is assumed that the emission profiles of $SO_2$ in 2013 are not significantly different from 2010. Stacks from the Imperial Oil Kearl facility are not in the CEMA inventory because those stacks started operation later than 2010. A comparison of observed plume locations, as outlined below in Section 2.7, demonstrates that the Kearl and Firebag stacks produce no discernable $SO_2$ plumes. Based on this comparison, there are 7 stacks which emit significant (more than 0.050 kg s$^{-1}$) $SO_2$. The 12 non-$SO_2$ emitting stacks all report less than 0.005 kg s$^{-1}$.

A flaring stack at the CNRL facility was added to the list (CNRL2) because daily reports indicated a large amount of $SO_2$ emissions were released from the flaring stacks for a one-week

period during the field study. However, by their nature (a high temperature flame at the top of the stack is used to loft pollutants upwards), CEMS monitoring of flare stacks is not possible with current technology, and hence emissions rates and stack parameters for this source are engineering estimates. The stack parameters for this flaring stack were parameterized using effluent velocity and temperature based on annual NPRI inventory values (NPRI ID 23275; NPRI Website, see ECCC & AEP, 2016).

Although NRPI data are available for the CNRL flaring stack, the other CNRL stack used here (a "sulphur recovery unit") has both CEMS and NPRI data available. This allows for a test of the variability in $T_s$ and $w_s$ through comparison of NPRI data (where annual average values are reported) and CEMS data (hourly) for this period and stack. For stack CNRL1 the annual average NPRI values were $T_s = 811$ K and $w_s = 17$ m s$^{-1}$, and the CEMS data averages for the study period are $T_s = 851$ K and $w_s = 4.1$ m s$^{-1}$ (a 5% temperature difference and more than a factor of 4 difference in flow rate). Hence there may be significant differences between data reported through both methods; by extension the CNRL2 values (for the one-week period it is active) should be considered only approximations.

All 8 stacks are listed in Table 1 and the locations of these 8 stacks are shown in Fig. 1. For comparison, average effluent velocities (calculated from flow rate and stack diameter as $w_s = 4V/\pi d_s^2$) and temperatures were calculated for each stack over the 84 hours of research aircraft flight time (with the exception of CNRL2, which is based on annual NPRI inventory values). These averages are shown for comparison only; plume rise in the analysis which follows is calculated using hourly CEMS data concurrent with the time of plume observations. Plume observations and the aircraft flight campaign are discussed in more detail in the following sections.

Table 1. CEMS stack parameters for all stacks within the flight area which emit significant SO$_2$, including location and elevation at the stack base ($z_{surface}$), stack height ($h_s$), stack diameter ($d_s$), effluent velocity at the stack exit ($w_s$), and effluent temperature at the stack exit ($T_s$). Velocities and temperatures shown here are averages for the entire flight period. Hourly CEMS values are used for plume rise calculations. Stack numbers (#) are for identification within this analysis and do not represent official reporting ID. The SO$_2$ emission rates from 2010 inventory are shown for comparison.

| Facility | # | Latitude | Longitude | $z_{surface}$ [m amsl] | $h_s$ [m] | $d_s$ [m] | $w_s$ [m/s] | $T_s$ [K] | SO$_2$ [kg/s] |
|---|---|---|---|---|---|---|---|---|---|
| Suncor | 1 | 57.0020 | -111.4770 | 257 | 106.7 | 5.8 | <0.1 | 404.3 | 0.14 |
| Suncor | 2 | 57.0050 | -111.4770 | 254 | 106.7 | 2.0 | 9.3 | 711.5 | 0.06 |
| Suncor | 3 | 57.0030 | -111.4770 | 256 | 137.2 | 7.0 | <0.1 | 336.3 | 0.19 |
| Suncor | 4 | 57.0060 | -111.4790 | 255 | 106.1 | 3.4 | 4.2 | 947.3 | 0.17 |
| Syncrude | 1 | 57.0410 | -111.6160 | 304 | 183.0 | 7.9 | 12.0 | 472.9 | 2.27 |
| Syncrude | 2 | 57.0480 | -111.6130 | 305 | 76.2 | 6.6 | 10.1 | 350.7 | 0.12 |
| CNRL | 1 | 57.3390 | -111.7380 | 284 | 106.7 | 3.4 | 4.1 | 851.1 | 0.20 |
| CNRL[*] | 2 | 57.3390 | -111.7380 | 284 | 109.0 | 1.4 | 6.2 | 1273.1 | N/A[*] |

[*] The CNRL#2 flaring stack is added based on NPRI inventory and is assumed to emit significant SO$_2$ for a 1-week period during the field study.

The relatively high flow rates and diameters of some stacks may lead to plume rise due to momentum alone, especially under stable conditions.  Briggs also developed similar equations for rise due to momentum (c.f. Briggs, 1984).  These equations are typically used when $F_b = 0$, and the plume is assumed to be either a vertical jet (momentum driven) or a bent over plume (buoyancy driven).  The potential effect of momentum on the plume rise is discussed in Section 4.4.

### 2.4 Measurement Platforms

Wind speed ($U$), wind direction ($\theta$), and temperature ($T_a$) data at the stack height and at the surface were estimated based on measurements made at either: one of two meteorological towers in the study region (WBEA: AMS03 and AMS05); or a radio-acoustic sounding system (*wind*RASS, Scintec).  Figure 1 demonstrates the sites of the WBEA meteorological towers, and the radio-acoustic sounding system (RASS).

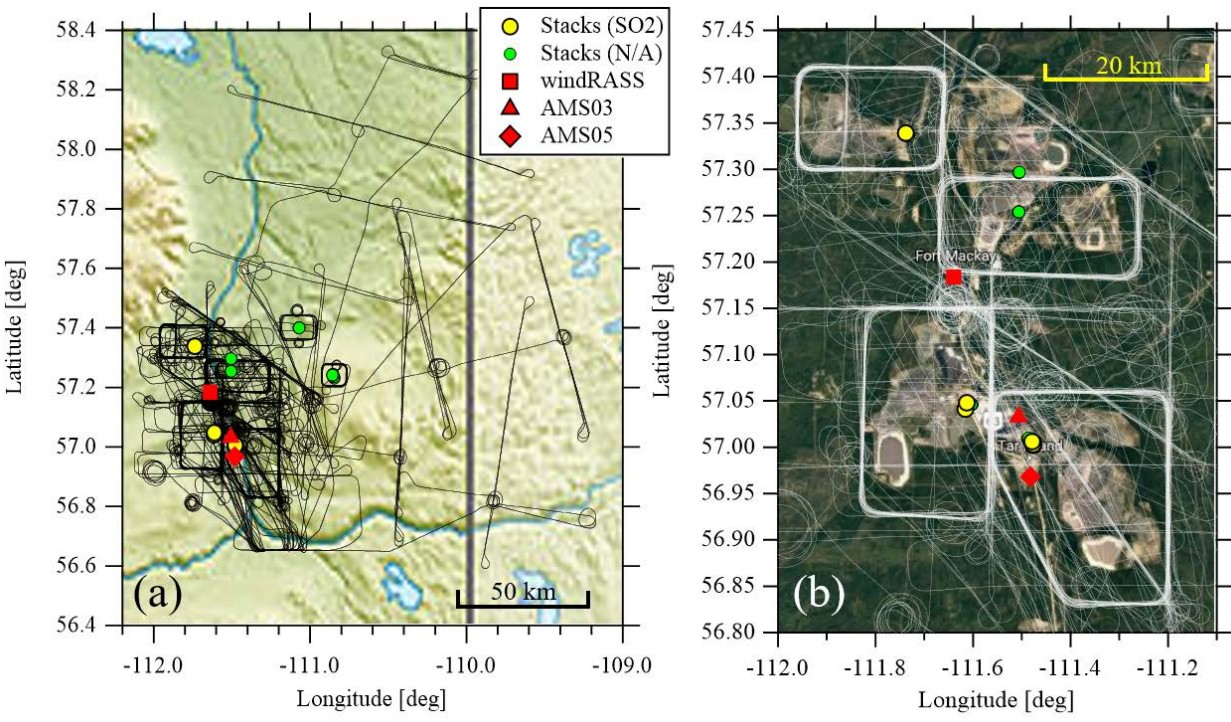

Figure 1. The flight tracks (black lines on (a), white lines on (b)) during the 22 flights of the JOSM study, compared to the location of: the facility stacks, including SO₂ emitting stacks used for this analysis (yellow circles) and non-SO₂ emitting stacks (green circles); the radio-acoustic profiler (windRASS, red square); and the WBEA meteorological towers, AMS03 (red triangle); and AMS05 (red diamond).  Stack towers in close proximity are overlapping.  The relief map (a) shows the extent of the flight area and the Athabasca river valley with the Alberta/Saskatchewan border shown at –110º longitude (Wikipedia, credit: Carport). The satellite image (b) is a close up in the region of the facilities (Google: Landsat/Copernicus, 2017).

The AMS03 tower measures wind speed, wind direction and temperature at heights of 20, 45, 100, and 167 m (all heights above ground level). The AMS05 tower measures wind speed and direction at heights of 20, 45, 75, and 90 m and temperature at heights of 2, 20, 45, and 75 m. Tower measurements are reported as 1-hour averages. The RASS measures wind speed and temperature (among other variables) between a minimum height of 40 m and a maximum height

which varies depending on wind conditions (Cuxart *et al.*, 2012). During the aircraft flight period, the maximum RASS measurement height varied from 130 m to 800 m, with an average 336 m. The RASS measurements are 15-min averages.

As part of JOSM, aircraft-based measurements were made in the Athabasca oil sands region between August 13 and September 7, 2013. The project included 22 flights, which were flown

in some combination of either box formations (circumnavigating a facility at variable heights in order to determine facility pollutant emissions), screen formations (flown perpendicular to the plume centreline axis to characterize the transformation of the plumes), spiral ascent and descent (to characterize boundary-layer structure), or horizontal area coverage (to verify satellite observations over a larger spatial extent). Figure 1 shows all these flight formations. Within the

22 flights, there were 16 box-flight formations and 21 screens used for this analysis. Aircraft flight times varied from approximately 2.5 hours to over 5 hours, typically in the mid-afternoon, for a total of 84 hours. Wind speeds and temperatures were measured from the aircraft with a Rosemount 858 probe, sampled at 32 Hz and averaged to 1 Hz. For details of the aircraft measurements, see Li *et al.* (2017), Liggio *et al.* (2016), and Gordon *et al.* (2015). The aircraft

flew at a minimum height of 150 m above ground level (agl). The maximum height of box formations varied from 500 m agl to 1300 m agl, while the maximum height of screen formations ranged from 350 m agl to 2000 m agl.

Table 2. Correlation coefficient ($R^2$) of wind speeds ($U$), wind directions ($\Psi$), and temperature ($T$) at

305 given comparison heights.

| | | Comparison | $R^2$ | | |
|---|---|---|---|---|---|
| | | Height | $U$ | $\Psi$ | $T$ |
| RASS | AMS03 | 167 m | 0.61 | 0.88 | 0.84 |
| AMS03 | AMS05 | 90 m | 0.80 | 0.94 | 0.98 |
| AMS05 | RASS | 90 m | 0.56 | 0.84 | 0.82 |
| Aircraft | RASS | < 200 m | 0.66 | 0.60 | 0.82 |
| Aircraft | AMS03 | < 200 m | 0.61 | 0.63 | 0.78 |

Tower, RASS, and aircraft measurements were compared over the 84 flight hours. The RASS was not operational until Aug. 17 (thus missing 3 flights); hence RASS data are compared for a reduced period. For comparison to the tower measurements, the 15-min RASS and 1-s aircraft

measurements were averaged to concurrent 1-hour values. For comparison to the RASS, the 1-s aircraft measurements were averaged to 15-min values. The resulting correlation coefficients are listed in Table 2. The aircraft wind and temperature measurements are also compared with the highest tower (AMS03) and the RASS. For comparison to aircraft measurements, the RASS

measurements at a height of 90 m were compared to all concurrent aircraft measurements below
200 m.  In the case of AMS03, the measurement at a height of 167 m was compared to all
concurrent aircraft measurements below 200 m.  The wind speed comparisons are best between
the two towers ($R^2 = 0.80$).  Wind direction compares well for the towers and the RASS ($R^2 >$
0.84).  Temperature compares well for all measurement platforms ($R^2 > 0.78$).  Generally,
comparisons with the aircraft give the lowest correlation values.  We note that the correlations of
Table 2 do not show potential local offsets in magnitude, and that the aircraft observations are
averages over a larger region which may not be spatially co-located with the towers.  We also
note from Figure 1(b) that towers AMS03 and AMS05 are less than 10 km apart, while the
RASS is approximately 20 km from the two towers.  The correlations between AMS03 and
AMS05 are higher than between either of these towers and the more distant RASS, and that
correlations with the aircraft have the lowest values, implying that some of the lower correlations
may reflect local heterogeneity in meteorological conditions.

We note that the Athabasca oil sands region is centered on the Athabasca River valley, with over
500m of vertical relief within 60 km of the facilities; the flow within the valley may be complex,
with frequent observations of shear between plumes from stacks at different elevations under
stable conditions.  The low correlations between the stations and between the stations and the
aircraft reflect this variation in local meteorological conditions.  We examine this possibility
through the use of a high resolution GEM-MACH simulation in our companion paper
(Akingunola et al, 2018).

## 2.5  Stability ($z/L$), Boundary-Layer Height ($H$), and Friction Velocity ($u_*$)

Stability, boundary-layer height, and friction velocity were all determined from the observations
using wind speed and temperature profiles from multiple height measurements.  The towers,
which have anemometers and temperature sensors at variable heights between 2 m and 167 m,
measured within the surface layer and are best suited for these estimations.  The RASS, which
has a minimum measurement height of 40 m, may not capture the surface layer effectively.  As
the aircraft did not fly below a height of 150 m, aircraft-based measurements cannot be used to
estimate the stability, boundary-layer height, and friction velocity.  For our analysis, we calculate
$L$, $H$, and $u_*$ to drive the Briggs parameterization (Eqns. 1-6) using observations from the two
towers (AMS03 and AMS05) and the RASS.

The atmospheric stability is determined using the Bulk Richardson Number, which is defined
(Garratt, 1994) as

$$R_i = \frac{gz_h}{\theta} \frac{\Delta\theta}{\Delta U^2}. \tag{8}$$

Here $\Delta\theta$ and $\Delta U$ are the potential temperature and wind speed differences over the height range
($z_h$).  The height range is determined as the difference in height between the highest
measurement location and the lower measurement location.  For example, $z_h =$147 m for
AMS03, $z_h = 55$ m for AMS05, and $z_h$ is variable for the RASS. The Richardson number is then
related to the stability parameter (Kaimal and Finnigan, 1994) as

$$\frac{z}{L} = \begin{cases} R_i & \text{for } R_i < 0 \\ \dfrac{R_i}{1 - R_i/R_{ic}} & \text{for } 0 < R_i < R_{ic} \\ +\infty & \text{for } R_i > R_{ic} \end{cases} \tag{9}$$

Here $R_{ic} = 0.25$ is the critical Richardson number, chosen as the mid-range of reported values (0.2, 0.25, or 0.5; Mahrt, 1981). For $R_i > R_{ic}$ there is no solution, so this is modelled as extremely stable boundary-layer with $L$ slightly larger than zero (to satisfy the stability condition $L > 0$). The Obukhov length is calculated from the stability parameter as $L = z_{max}/(z/L)$, where $z_{max}$ is the highest measurement height of 167 m, 90 m, or up to 800 m for AMS03, AMS05, and the RASS respectively.

Boundary-layer height can be parameterized for stable and unstable conditions following Mahrt (1981) as

$$H = \frac{R_i T_{sur}}{g} \frac{U(H)^2}{\theta(H) - \theta_{surface}}, \tag{10}$$

where $R_i$ is the bulk Richardson number and $U(H)$ and $\theta(H)$ are the respective wind speed and potential temperature at the boundary-layer height and $\theta_{surface}$ is the potential temperature at the surface. Since measurements at the boundary layer height may not be available, we approximate the wind speed to temperature gradient ratio in Eq. 10 as $U(z_{max})^2/(\theta(z_{max}) - \theta_{surface})$.

The boundary-layer height derived from Eq. 10 can be compared to the boundary-layer height estimated from in-situ aircraft measurements of the $CH_4$ mixing ratio during vertical profile flight formations. These $CH_4$ profiles demonstrate a well-defined background level above a given height, with elevated $CH_4$ mixing ratios below this height. The boundary-layer heights determined by the aircraft measurements range from 340 m to 1790 m with an average of 1180 m. The values of $H$ derived from Eq. 10 using the AMS03 tower data for the same time periods as the flights range from 460 m to 3050 m, with an average of 1160 m.

The friction velocity ($u_*$) was determined from the wind speed profile (Garratt, 1994) as

$$u(z) = \frac{u_*}{k}\left[\ln\left(\frac{z}{z_o}\right) - \Phi\right], \tag{11}$$

where $z_o$ is the roughness length, $k = 0.4$, and the stability parameter is

$$\Phi = \begin{cases} 2\ln\left(\frac{1}{2}(1 + x_o)\right) + \ln\left(\frac{1}{2}(1 + x_o^2)\right) - 2\operatorname{atan}(x_o) + \frac{\pi}{2}, & \text{for } \frac{z}{L} < 0 \\ -5\frac{z}{L} & \text{for } \frac{z}{L} > 0 \end{cases} \tag{12}$$

with $x_o = (1 - 16z/L)^{1/4}$. A least-squares method is used for each hourly profile to determine an appropriate $z_o$ for the measurement location, which is taken as the median value of all the hourly fits. This median $z_o$ value calculated using this method varies considerably by location (1.5m for AMS03, 0.75 m for AMS05, 10.1 m for RASS). The median $z_o$ values were then used to calculate $u_*$ using the hourly wind speed measured at the highest location. The calculation of $u_*$ with the RASS may be inaccurate due to the lack of measurements between the surface and a height of 40 m. However, the large difference in values of $z_o$ may be also due to the different

environment surrounding the measurement locations, since the towers are surrounded by forest and the RASS is located in the town of Fort McKay.

It is noted that parameterizing stability without a measurement of heat flux and estimating boundary-layer height based on near surface measurements may lead to significant uncertainties in these values. This will also affect the estimation of $u_*$, and may be evident in the median $z_o$ values for the RASS, which are very large even for a town with 2 or 3-story buildings. Tests to determine the sensitivity of the calculated plume rise to these variables ($L, H, u_*$) are discussed in Section 4.2.

## 2.6 Stability Profile Measurements for the Layered Method

To drive the layered method discussed in Section 2.2, profiles of temperature and wind speed were derived for each box and each screen using RASS and aircraft observations. RASS layers were 10 m thick to match the instruments resolution. The lowest RASS measurement is at a height of 40 m, well below the lowest stack height (76 m). Because the maximum observation height of the RASS varies (with an average of 336 m), it was necessary to extrapolate temperature and wind speed above the maximum measurement height in some cases. This was done by assuming a constant wind speed and a constant temperature gradient, based on measurements in the highest 100 m of observations.

For aircraft observations, the box and screen flights were designed to approximate 100 m vertical spacing between each box circuit or screen pass. Based on this resolution we use a layer thickness of 100 m for the layered method driven by aircraft observations. Testing demonstrates that the algorithm is not sensitive to the layer thickness. Flight measurements of wind ($U$) and temperature ($T$) for each box and screen are averaged in vertical layers within the 100 m spacing. Since there are no measurements below a height of 150 m agl, the temperature at the lowest layer ($0 < z < 100$ m) is extrapolated by assuming a constant lapse rate and stability below 200 m (i.e. $S_{j=1} = S_{j=0}$). There are no cases of calculated plume height based on the layered method exceeding the maximum aircraft measurement height and hence no need for upward extrapolation of the measurements.

Our temperature profiles for the layered method thus have as key assumptions: (1) that the profiles at the RASS location and derived from the aircraft are representative of conditions at the stacks, and (2) that the extrapolations and vertical resolution used here provide a reasonable representation of the atmospheric temperature profile.

## 2.7 Measured Plume Heights and Stack to Plume Matching Algorithm

The aircraft measured numerous pollutants, of which $SO_2$ is used here to define the stack plume locations since approximately 95% of the $SO_2$ emissions in the region originate in stacks (Zhang *et al.*, this issue). The $SO_2$ analyzer (Thermo Fisher Scientific, model 43i) on the aircraft measured at a rate of 1 Hz. The flight paths were designed to create a 100 m spacing between measurement points (in both horizontal, $s$, and vertical, $z$) in order to optimize interpolation of

the measurements. The measurements were interpolated in *s* and *z* using simple kriging as outlined in the Topdown Emission Rate Retrieval Algorithm (TERRA; Gordon *et al*., 2015). This creates two-dimensional images of $SO_2$ mixing ratio. For box flights, which circumnavigate the facilities, the *s* coordinate is the distance along the box in the counter-clockwise direction from the southeast corner. For screens, *s* is the lateral distance along the screen, generally perpendicular to the wind direction. Below the lowest flight path (at 150 m agl), no interpolation is performed and the screen is left blank between this level and the ground. Figures 2 and 3 show example box and screen flight paths in both horizontal (Fig. 2) and vertical (Fig. 3) profiles.

A semi-empirical approach was used to match each stack to the observed plume locations. The wind direction measured from the aircraft was averaged for the duration of each box or screen. Tower or RASS-based wind direction measurements were not used, as an initial comparison of wind directions and observed plume locations demonstrated that the aircraft measurements are a better representation of the wind direction associated with plume transport than surface
measurements. This agreement is most likely due to the consistent proximity of the aircraft to the stack sources; the towers and RASS locations can often be much further away (Fig. 1).

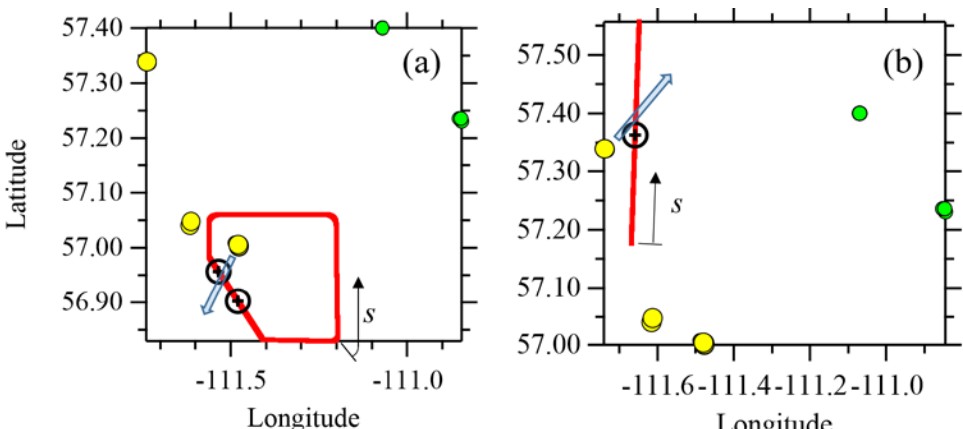

Figure 2. Example horizontal flight path of a box flight (a) and a screen flight (b). Flight paths for the box and screen portion of the flight shown as red lines. Stack locations are shown as filled yellow circles ($SO_2$ emitting) and green circles (non-$SO_2$ emitting). The blue arrow shows the forward trajectory of the plume using the average wind direction during each flight segment. The plume locations determined by observations (Fig. 3) are shown as black cross-hairs on the flight paths. The location of the flight path
coordinate *s* origin is labeled in each figure.

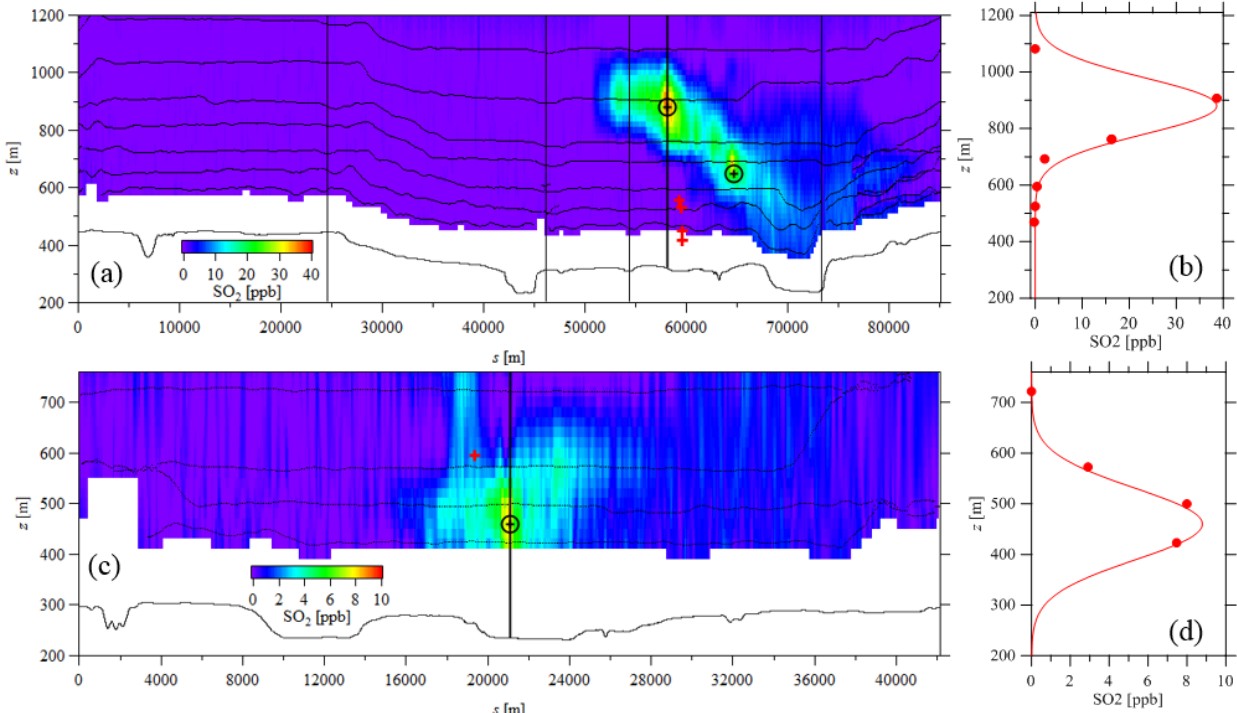

Figure 3. The interpolated images for the box flight (a) and the screen flight (c) (as Fig. 2). The aircraft flight paths are marked by the finely spaced (1 Hz) black dots. The surface location ($z_{surface}$) is shown below the flight path. Interpolation is removed between the lowest flight path and the surface, following the TERRA method. In the box (a), the thin vertical lines show the box corners (see Fig. 2a). The plume locations determined by the Briggs plume rise and the forward trajectories ($s_{int}, z_h$) are marked by red plus signs. The plume locations determined by observations ($s_p, z_p$) are shown as black cross-hairs. The Gaussian fitting used to improve plume height estimation is demonstrated (b,d) for the location marked by the thick vertical black line in each image.

The average wind directions were then used to predict the direction of plume transport downwind of each stack. The intercept of each plume's predicted path with the box or screen ($s_{int}$) was calculated based on this forward trajectory from the stack source to the box or screen intercept. Example box and screen flight paths, forward trajectories, and observed plume locations are shown in Figure 2 for the flights on Aug. 29 (Fig. 2a) and Aug. 15 (Fig. 2b). This simple forward trajectory methodology ignores the local effects of topography, vertical winds, and the variability of the wind during the box or screen segment of each flight (typically less than 2 hours of flight time). Some screens were flown up to 150 km from the 8 stacks (see Fig. 1). Since other stratification, topography, and diffusion effects may influence a plume height at such a large distance from the plume origin, we restrict our analysis to box walls and screens within 50 km of the plume stack sources.

Plume rise ($\Delta h$) was calculated for each stack based on the Briggs parameterization, the observed meteorological conditions at the tower or RASS locations (or RASS and aircraft data, for the layered approach), and the CEMS stack parameters, all averaged for the duration of the box or screen flight periods. This calculation also defined the estimated plume centreline location at

each box or screen as $(s_{int}, z_h)$, where $z_h = z_{surface} + h_s + \Delta h$ and $z_{surface}$ is the surface elevation (amsl) at the intercept.

The flight path observations are converted to two-dimensional $(s, z)$ images by kriging interpolation following the method outlined in Gordon *et al.* (2015). Example interpolated images from both a box and a screen flight are shown in Figure 3. A disadvantage of kriging
interpolation of the aircraft data is that the maxima of the plumes will always be fixed at a flight measurement location. To improve the resolution of observed plume height from the interpolated images, the aircraft measurements within a 100-m wide window (i.e. $s \pm 50m$) are fitted to a Gaussian vertical profile. Example profiles are shown in Figures 3b and 3d, which correspond to the windows shown as thick black lines through the maximum SO$_2$ locations (the
plume centres) in Figures 3a and 3c. The maxima of the Gaussian fits for each identified plume are then used to identify the prominent plume locations as $(s_p, z_p)$. The identified plume locations are visually compared to the predicted Briggs plume locations based on the forward trajectories for each box or screen $(s_{int}, z_h)$.

Non-stationarity of the wind speed, wind direction, and plume buoyancy during the
measurements is a potential source of uncertainty as each flight circuit (or pass) around the facility can take between 10 and 15 minutes. This effect is discussed in Gordon et al. (2015) for this flight campaign. Although this can have significant effect on the calculation of emissions, the effect on the estimation of plume height should be less than the vertical distance between passes (~100 m). Further, some flights were flown from bottom to top, while others were from
top to bottom, so there should be no directional bias on average.

Each calculated plume location $(s_{int}, z_h)$ was paired with each nearby observed plume location $(s_p, z_p)$ to maximize the correlation of calculated and observed plume heights. For example, the calculated plume rise from three stacks would be paired with three observed plume heights by matching the lowest calculated plume height to the lower observed plume height; the middle
calculated plume height to the middle observed plume height; and the highest calculated plume height to the highest observed plume height. This gave the highest correlation between predicted values and observations. For a single plume observation and multiple SO$_2$-emitting upwind stacks, the stack plumes were assumed to have merged and the calculated plume height for each stack was paired to the same observed plume height. The merging of plumes is supported by
visual observation by the authors during the field study, especially far downwind of the stack locations.

For the example of the Aug. 15 screen flight (Fig. 2b and Figs. 3c,d), the forward trajectory and Briggs algorithm model intercept the flight screen approximately 2 km further south, and 140 m higher, than the observed plume centre, indicating the possibility of more complex wind flow
than a simple trajectory. In the example of the Aug. 29 box flight (Fig. 2a and Fig. 3a,b), there are two observed plumes along the NW-SE oriented wall of the box. The forward trajectory model places the plume intercept between these two plumes, closer to the vertically higher and more northern observed plume at the horizontal location given by $s = 58$ km. There are four stacks within the box, two of which have calculated intercept heights near $z_h = 540$ m and two
of which have calculated intercept heights near $z_h = 430$ m. All four calculated values are

clearly well below the observed intercept heights ($z_p = 650$ m and 880 m). This demonstrates some ambiguity and subjectivity in this analysis, as four calculated plume locations must be matched to two observed plumes. As described above and for the purposes of statistical comparisons, we match the highest two modeled plumes (near heights of 540 m) with the highest observed plume (880 m) and the lower two modeled plumes (near heights of 430 m) with the lower observed plume (650 m).

## 3. Results

**3.1 Comparison of Measurement Platforms**

The topography of the Athabasca oil sands region can be generally described as a north-south river valley approximately 1 to 5 km in width, within a larger and more gradually sloped north-south valley between 10 and 50 km in width, and up to 500 m of vertical relief (Fig. 1a). Local surface wind patterns can be heterogeneous, especially within the valley. The AMS03 and 530 AMS05 towers are in the vicinity of the Suncor stacks and the Syncrude stacks (Table 1), while the RASS is nearly equidistant to the 8 stacks used for this analysis (Fig. 1b).

As an approximate measure of the uncertainty associated with local meteorology, plume rise values from the 8 stacks are compared using the Briggs parameterization (Eqs. 1-6) with all 3 meteorological measurement platforms (i.e. AMS03, AMS05, and RASS) as well using the 535 layered method (Eq. 7) with both RASS and aircraft measurements. This comparison was done for all concurrent times during which the aircraft was flying box or screen patterns. There were approximately 26 hours during which the aircraft flew in a box pattern and 20 hours during which the aircraft flew in a screen formation, for a total of more than 46 hours. The resulting distributions of calculated plume heights for these 46 hours of flight time for the 8 stacks are 540 compared in Figure 4.

The distributions of plume rise heights are similar for the Briggs parameterization with the three fixed, near-surface measurement platforms. Approximately 90% of the plume rise values calculated with the AMS tower and RASS measurements are below approximately 250 m, with half or more below 75 m. With the layered method, the plume heights calculated with the RASS 545 measurements are similar to those calculated with aircraft measurements. As with the Briggs parameterization, approximately 90% of the plume rise values are below 250 m; however, more than half of the plume rise heights calculated with the layered method are above 125 m.

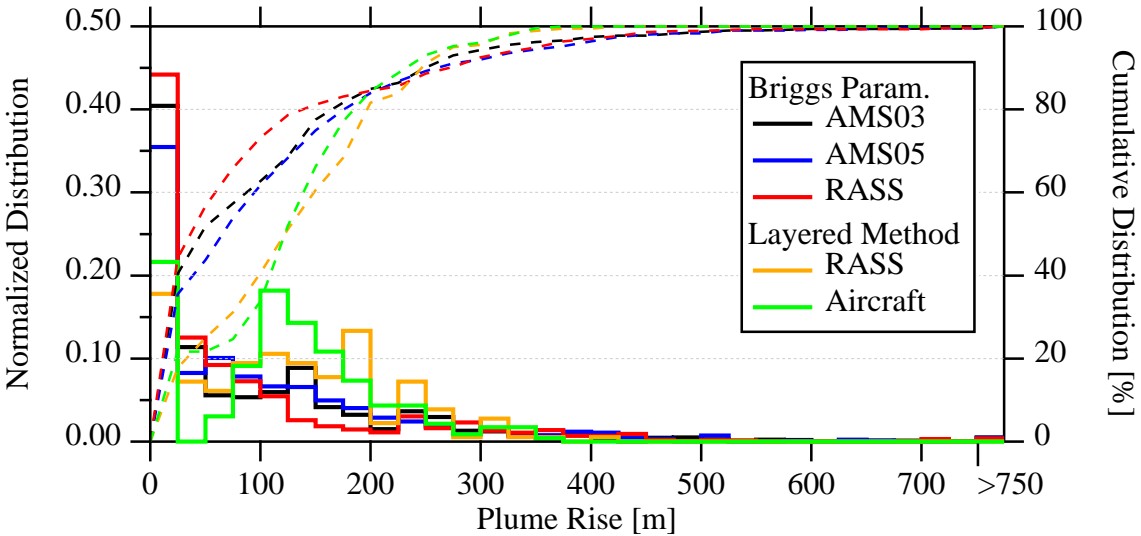

Figure 4. The distribution of calculated plume rise (Δh) using Briggs parameterization (Eqns. 1-6) with
input data from the AMS03 and AMS05 towers and the RASS profiler, and the layered method (Eq. 7)
with input data from the RASS profiler and the aircraft. Distributions are shown for each hour (using the
46 hours of box and screen flight times) and for all of 8 SO₂ emitting stacks combined. The right-most
histogram bin is the sum of all values of Δh > 750 m. Cumulative distributions shown by dashed lines.

## 3.2 Predicted Plume Rise

The plume rise was calculated for each flight for each stack with the Briggs parameterization for
each input (towers, RASS) as well as with the layered method (RASS, aircraft). These plume
rises were then paired with the measured plume locations following the method described in
Section 2.7. For simplicity, the parameterized plume rise is described as $h_B = \Delta h$, and the
observed plume rise is described as $h_M = z_p - z_{surface} - h_s$. Results of this comparison are
shown in Figure 5. The analysis resulted in 82 stack-to-observed plume pairings, for each
measurement platform. (Note that a smaller number of pairings were possible for the RASS,
which was not in operation for 4 of the 22 flight days). Table 3 compares the results for each
measurement method. The low slopes ($b$ <0.5), significant intercepts ($44 < a < 107$ m), and
low correlation coefficients ($r^2 \leq 0.2$) demonstrate that the Briggs parameterization of plume
rise was a poor predictor of actual plume rise. For a 95% confidence (calculated from the
standard error of the slopes) none of these slopes is significantly different from zero.

Using the tower or RASS measurements with the standard Briggs parameterization suggests an
average underestimation (based on the average ratio) between 18% (RASS) and 45% (AMS03).
The layered method using the RASS and aircraft-based measurements predicts a plume rise that
is, on average, nearly half (47 – 49%) of the observed value. In all cases, more than half of the
plume rise values are underestimated by more than a factor of 2, and between 22 to 42% of
predicted plume rise values are within a factor of 2 of the observations.

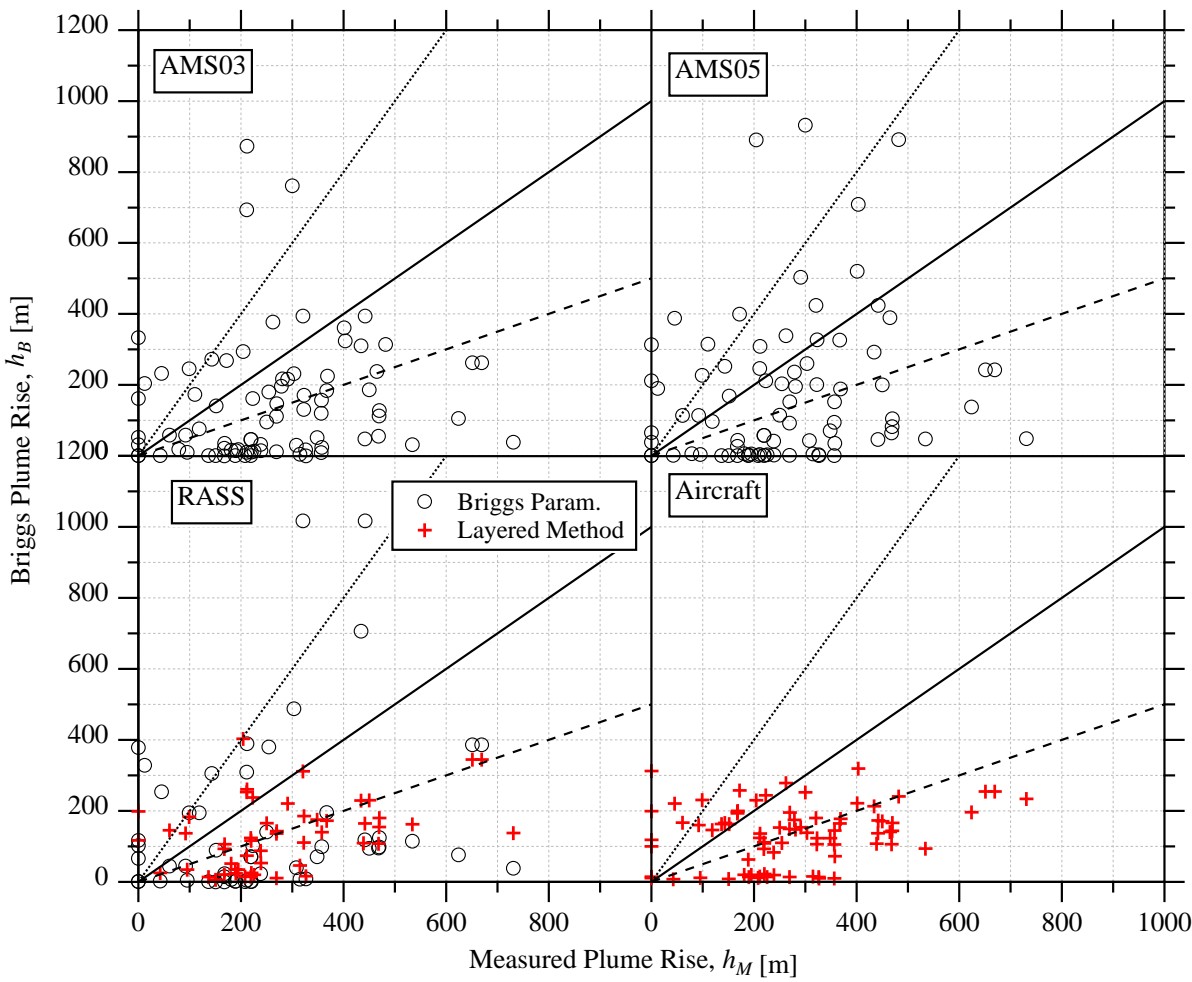

Figure 5. Comparison of the predicted plume rise from the Briggs parameterization used in GEM-MACH with the measured plume rise as determined by various atmospheric measurements described in the text. Black circles indicate the Briggs parameterization (Eqns. 1-6) and red crosses indicate the layered method (Eq. 7). Lines demonstrate 2:1 (dotted), 1:1 (solid), and 1:2 (dashed) ratios for comparison.

Table 3. Statistics comparing the predicted to measured plume rises using both the Briggs parameterization (Eqns. 1-6) and the layered method (Eq. 7). The intercept ($a$) and slope ($b$) of least-squares fit, average calculated ($\overline{h_B}$) and observed ($\overline{h_M}$) plume rises, ratio of all values $\overline{h_B}/\overline{h_M}$, correlation coefficient ($r^2$), fraction of individual ratios of $h_{B,i}:h_{M,i}$ below the 1:2 ratio (<0.5), within a factor of 2 (>0.5 & <2), and above the 2:1 ratio (>2), and the number ($n$) of plume to stack matches used for each comparison.

| | $a$ [m] | $b$ | $r^2$ | $\overline{h_B}$ [m] | $\overline{h_M}$ [m] | $\overline{h_B}/\overline{h_M}$ | Ratio <0.5 | >0.5 & <2 | Ratio >2 | $n$ |
|---|---|---|---|---|---|---|---|---|---|---|
| | Briggs Parameterization, Buoyancy Rise Only | | | | | | | | | |
| AMS03 | 104 | 0.16 | 0.02 | 145 | 263 | 0.55 | 54% | 32% | 15% | 82 |
| AMS05 | 107 | 0.25 | 0.04 | 173 | 263 | 0.66 | 52% | 32% | 16% | 82 |
| RASS | 78 | 0.51 | 0.07 | 207 | 254 | 0.82 | 55% | 22% | 22% | 58 |
| | Layered Method | | | | | | | | | |
| RASS | 63 | 0.24 | 0.16 | 130 | 275 | 0.47 | 53% | 42% | 6% | 53 |
| Aircraft | 100 | 0.13 | 0.06 | 134 | 272 | 0.49 | 57% | 32% | 11% | 79 |

## 4. Discussion

### 4.1 Stability Classification

Table 4 lists the frequency of each stability class during box and screen flight times according to each measurement platform as determined by the sign and magnitude of the Obukhov length ($L$). Stable classification is separated as either due to small positive values of $0 < L < 2h_h$, or stack height above the boundary layer height ($h_s > H$). The RASS and the two towers give similar, predominantly (70 to 94%) neutral, stability during the flights, with RASS indicating the highest frequency (94%) of neutral conditions. Of these three measurement platforms, only the measurements of AMS05 predict plume rise through unstable conditions. We also note that AMS03 and AMS05 are in close spatial proximity to each other (less than 10km), suggesting substantial local changes in stability, again arguing for heterogeneity in the local conditions.

Based on previous studies summarized in VDI (1985), the authors suggested a reduction of the Briggs parameterization by 30% in neutral conditions. Although the atmospheric stability is predominantly classified as neutral in our analysis, we are seeing an underestimation by the Briggs parameterization, in contrast to the previous studies.

Stability was determined using the RASS and aircraft temperature profile measurements based on a comparison of the temperature profile to the adiabatic lapse rate ($\Gamma = g/c_p = 0.0098$ K/m). The temperature profiles were derived from measurements between the minimum aircraft height of 150 m and 300 m (agl). The profile was considered neutral if $-dT/dz$ was within 20% of $\Gamma$. Because the RASS profiles demonstrated very different lapse rates near the surface compared to further aloft, these data were separated into near-surface (<100m) and higher (>100m). The profile measurements used for the layered method give a much different indication of stability class, with predominantly stable conditions between for 53% and 89% of the time. The RASS

measurement profiles demonstrate a higher frequency of stable conditions near the surface
(based on comparison to the lapse rate). For the RASS measurements, there is a significant
difference between stability classifications based on Obukhov length compared to stability
classifications based on the temperature lapse rate, suggesting that either these two methods are
not directly comparable, or that significant spatial heterogeneity exists within the region (as is
also implied by the comparison in stability classes noted at AMS03 and AMS05). The layered
approach of Eq. 7 is based on the assumption of neutral or stable conditions. For unstable
conditions we follow the assumptions outlined in Akingunola et al. (2018) and assume $S = 0$.
Since there is a relatively low frequency of unstable conditions in all cases (4% to 13%), any
error caused by the assumption of $S = 0$ during unstable conditions is likely small.

Table 4. Frequency of each stability type during flight times determined by each measurement platform.
Stability is either determined by parameterization of Obukhov length ($L$, see Section 2.1), by comparison
of the temperature profile with the dry adiabatic lapse rate ($\Gamma$), or using the Pasquill-Gifford stability
classification scheme (P.-G.).

| | Basis | Unstable | Neutral | Stable $(h_s < H)$ | Stable $(h_s > H)$ |
|---|---|---|---|---|---|
| AMS03 | $L$ | 0% | 70% | 12% | 18% |
| AMS05 | $L$ | 26% | 66% | 2% | 6% |
| RASS | $L$ | 0% | 94% | 0% | 6% |
| RASS (<100m) | $\Gamma$ | 4% | 7% | 89% | |
| RASS (>100m) | $\Gamma$ | 13% | 33% | 53% | |
| Aircraft (>150m) | $\Gamma$ | 8% | 23% | 69% | |
| AMS03 | P.-G. | 45% | 55% | 0% | |
| AMS05 | P.-G. | 60% | 40% | 0% | |

A comparison is also made using the Pasquill-Gifford (Turner and Schulze, 2007) stability class,
based on cloud cover and the wind speed at 10-m ($U_{10m}$). The P-G stability class specifies that
during moderate daytime radiation ("a summer day with few broken clouds with the sun 25-60°
above the horizon"), the atmosphere will be unstable (Classes A, B, or C) for wind speeds
$U_{10m} < 5$ m s$^{-1}$. For days with some cloud and $U_{10m} > 5$ m s$^{-1}$ or for completely overcast days,
the atmosphere will be neutral (Class D). According to the Pasquill-Gifford system, stable
conditions (Classes E, F) will only occur at night (all flights were during daylight hours). Here
$U_{10m}$ is determined from the lowest tower measurements (20 m) and Eq. 11, and cloud
conditions are estimated from photographs taken during the flights. This results in
predominantly unstable and neutral conditions, as shown in the first two rows of Table 4.

Hence all three methods produce a different prominent stability class: the Obukhov length
calculation predicts mostly neutral conditions; the lapse rate predicts mostly stable conditions;
and the Pasquill-Gifford stability classes predict an approximately equal occurrence of unstable
and neutral conditions. Both the Obukhov length and Pasquill-Gifford class approaches show a
substantial difference in the frequency of occurrence of unstable conditions between towers

AMS03 and AMS05, underscoring the local variability which may exist in temperature profiles. In light of this disagreement, we test the change in results with different stability classification schemes in Section 4.4 in order to estimate the extent to which the average plume rise depends on the stability classification.

**4.2 Sensitivity to Input Variables**

The above analysis suggests the potential for substantial variability between measurement locations, which may be due to heterogeneity of the terrain and surface conditions in the area. Here we perform a simple test of the sensitivity of the Briggs algorithm to uncertainties in input variables due to this variability between measurement platforms. Input variables are modified

based on differences between the AMS03 and AMS05 measurement platforms. First, the average plume rise is calculated for the box and screen flight times for the 8 stacks used in the analysis using AMS03 measurements as input. The input variables were then modified by the ratio of the average absolute difference between stations to the mean value (i.e. $\overline{|X_{03} - X_{05}|}/\overline{X}$, where $X_{03}$ and $X_{05}$ are the measurements variables at AMS03 and AMS05 towers respectively,

and $\overline{X}$ is the mean value from both stations combined). Instead of modifying the surface temperature ($T_{surface}$) directly, the difference between the air temperature at stack height and surface temperature ($\Delta T = T_a - T_{surface}$) is modified by a fraction, as it is the difference that drives the parameterization (through Eqns. 2, 8, and 10). The average plume rise was then recalculated with the modified variables to determine the resulting change in average plume rise

relative to the average plume rise calculated with unmodified input variables.

Table 5. Percent change in average plume height ($\Delta h((1 \pm R)X)/\Delta h(X)$), where $X$ is the modified parameter (i.e. $T_a, U$, etc.). $\overline{X}$ is the average of each variable from the two tower measurements (AMS03

and AMS05) and $\overline{\Delta X}$ is the average difference. The "Low" value is the average change in plume rise calculated with $(1 - R)X$ and the "High" value is the average change in plume rise calculated with $(1 + R)X$. All averages are for the 46 flight hours (box and screen flight times) and 8 stacks used in the analysis.

| Variable | Units | $\overline{X}$ | $\overline{\Delta X}$ | $R = \overline{\Delta X}/\overline{X}$ | Low | High |
|---|---|---|---|---|---|---|
| $T_a$ | K | 293.6 | 0.26 | 0.1% | 1.1% | -2.2% |
| $U$ | m/s | 5.1 | 0.70 | 14% | 23.1% | -15.6% |
| $\Delta T = T_a - T_{surface}$ | K | -1.4 | 0.45 | 31% | -3.9% | 3.4% |
| $H$ | m | 1150 | 990 | 71% | -27.0% | 6.7% |
| $u_*$ | m/s | 0.45 | 0.06 | 29% | 6.1% | -7.7% |
| $L$ | m | -132 | 90 | 165% | -14.9% | 0.3% |

Average percentage changes in the plume rise for each modification for each measurement platform are listed in Table 5. The largest differences between the two measurement locations are boundary-layer height ($H$, 71%) and Obukhov length ($L$, 165%). This is expected as the parameterizations of Eqns. 9 and 10 are known to be unreliable without heat-flux or upper air measurements. A decrease in boundary-layer height values by 71% leads to an average decrease in the plume rise of 27%, while an increase in boundary-layer height by 71% leads to an average increase in plume rise of 6.7%. Although the average difference in wind speeds between measurement stations is relatively low (14%), this has a considerable impact on the plume rise, ranging from a 23.1% increase to a 15.6% decrease in average plume rise. This is in contrast to air temperature ($T_a$), temperature difference ($\Delta T$), and friction velocity ($u_*$), which all results in an average change in plume rise of less than 8%.

The table identifies the variables with the largest impact on the parameterization results, hence which variables require the greatest accuracy when obtained from a meteorological model forecast. These results also help explain the low correlation coefficients of the observation-driven plume rise height comparisons (Table 3), as uncertainty in the estimation of these derived quantities will lead to uncertainty in individual plume rise estimations.

## 4.3 Horizontal Distance to Plume Rise

If the stacks are physically close enough to the interception of the plume with the box walls or screens it may be the case that the plumes have not travelled a sufficient distance to reach the maximum plume rise that is parameterized by the Briggs algorithms. Briggs (1984) also developed parameterizations of downwind distance to maximum plume rise. A plume in stable conditions will reach its final rise (Briggs, 1984) at

$$x_e = 4.7 \left( \frac{U}{\sqrt{S}} \right). \tag{13}$$

A plume in neutral conditions will reach its final rise (Briggs, 1975) at

$$x_e = \begin{cases} 49 F_b^{5/8} & \text{for } F_b < 55 \text{ m}^4\text{s}^{-3} \\ 119 F_b^{2/5} & \text{for } F_b > 55 \text{ m}^4\text{s}^{-3} \end{cases} \tag{14}$$

In unstable conditions, the plume fumigates and is evenly distributed in concentration between the surface and a height of $1.5\Delta h$, based on the assumption that the half-width of the plume is $0.5\Delta h$. Although no parameterization has been developed for the distance required to reach maximum plume rise in unstable conditions, Briggs (1984) provides a parameterization of the average horizontal distance to fumigation (contact of the plume with the surface) as

$$x_f = \frac{U}{w}(h_s + 0.5\Delta h), \tag{15}$$

where the average downdraft speed is $w = 0.8 u_*$, following Briggs (1984).

Using the AMS03 input data as an example, none of the 87 matched plumes have distance from stack to measurement location ($x_d$) less than the horizontal distance to reach maximum plume rise ($x_d < x_e$) in neutral or stable cases, and there are no unstable cases (Table 4). As discussed above, the analysis is limited to plume sources that are within 50 km of the box walls or screens.

The distances between stacks and box walls (following the forward trajectories) range from 4 to 16 km, while the distances between stacks and screens ranges from 3 km to more than 150 km. There are 8 screens located with within 40 km of the stack sources and 12 screens located more than 60 km of the stack sources (there are none in the 40 – 60 km range). Tests demonstrate (discussed in the next section) that including the 12 screen plume observations beyond 60 km from the sources in the analysis results in lower correlations and poorer performance of the Briggs parameterizations, as expected.

Given that the observed plume rise is generally much higher than the calculated plume rise, it should also be the case that distance to maximum plume rise is also underestimated. If it is assumed that the plume reaches its maximum height at the measurement location and the predicted plume rise ($h_B$) is less than the measured plume rise ($h_M$), then the actual distance to maximum plume rise can be estimated as $x_e' = x_e\, h_M/h_B$. Using this modified distance to plume rise, 13% of the plumes have distance to maximum plume rise greater than the distance between stack and screen (or box wall). This indicates that for these plumes, the assumption that $x_e' = x_d$ is incorrect and the maximum rise for these plumes is higher than $h_M$. Hence, the parameterized plume rise may underestimate the actual plume rise in some cases due to the measured plumes not reaching their maximum height. This magnitude of the underestimation is investigated as one of the modifications discussed below.

## 4.4 Modifications to the Plume Equations

To investigate the underestimation of plume rise by the parameterization, we recalculate the predicted plume rise with a number of modifications. For ease of comparison, we use only the AMS03 tower data to drive the algorithm. Table 6 lists the results of these modifications. The "base case" is the analysis as described in the preceding sections with no modifications. The "base case" statistics are reprinted in Table 6 (case 0) from the first line of Table 3 in order to facilitate comparison. The results are presented as scatter plots for each case (following Fig 5.) in the supplementary material. Each of the comparison studies presented as different cases in Table 6 are described in more detail in the sub-sections which follow.

## 4.4.1 Separation of Individual Stacks

Cases 1 through 8 in Table 6 provide statistics for the stack-plume matching separated by each of the 8 stacks as listed in Table 2. Half of the stacks demonstrate very strong underestimation of plume rise, with ratios of calculated to observed plume rise between 4% and 13%. In the cases of the Suncor stacks (1 and 3), these are large diameter stacks ($d_s = 5.8$ and 7.0 m, see Table 1) with very low effluent exit velocities. The average exit velocity of these stacks over the duration of the flights was $w_s < 0.1$ m s$^{-1}$ (Table 1). The CNRL stacks, by comparison, have relatively moderate and small diameters (3.4 m and 1.4 m) and moderate exit velocities (averages of 4.1 and 6.2 m s$^{-1}$ over the flight durations). This suggests that the underestimation of the plume height may result from either (inaccurately) low estimates of volume fluxes from these facilities, or that plume rise equations themselves are unsuitable for stacks with these conditions. This

does not appear to be the case for the CNRL stacks. However, there are only two stack-plume matches for each CNRL stack, so this is not a very statistically representative sample.

Table 6. Statistics comparing the predicted to measured plume rises using the Briggs parameterization (Eqns. 1-6) with either select conditions only or modification to the analysis. Cases are described in further detail in the text. Variables are defined as in Table 3.

| Case | # | $a$ [m] | $b$ | $r^2$ | $\overline{h_B}$ [m] | $\overline{h_M}$ [m] | $\overline{h_B}/\overline{h_M}$ | Ratio < 0.5 | >0.5 & <2 | Ratio > 2 | $n$ |
|---|---|---|---|---|---|---|---|---|---|---|---|
| Base Case | 0 | 105 | 0.14 | 0.02 | 143 | 265 | 0.54 | 55% | 30% | 14% | 83 |
| Suncor 1 | 1 | 1 | 0.03 | 0.32 | 6 | 178 | 0.04 | 91% | 0% | 9% | 11 |
| Suncor 2 | 2 | 140 | -0.01 | 0.00 | 137 | 260 | 0.52 | 73% | 9% | 18% | 11 |
| Suncor 3 | 3 | 8 | 0.00 | 0.00 | 9 | 199 | 0.04 | 92% | 0% | 8% | 12 |
| Suncor 4 | 4 | 235 | -0.21 | 0.02 | 175 | 286 | 0.61 | 50% | 33% | 17% | 12 |
| Syncrude 1 | 5 | 289 | 0.02 | 0.00 | 294 | 296 | 1.00 | 18% | 53% | 29% | 17 |
| Syncrude 2 | 6 | 149 | 0.12 | 0.04 | 185 | 298 | 0.62 | 25% | 69% | 6% | 16 |
| CNRL 1 | 7 | 66 | -0.04 | N/A | 49 | 395 | 0.13 | 100% | 0% | 0% | 2 |
| CNRL 2 (NPRI) | 8 | 100 | -0.23 | N/A | 15 | 374 | 0.04 | 100% | 0% | 0% | 2 |
| Neutral Cases Only | 9 | 101 | 0.13 | 0.01 | 134 | 244 | 0.55 | 56% | 26% | 18% | 50 |
| Stable Cases Only | 10 | 116 | 0.14 | 0.04 | 157 | 296 | 0.53 | 55% | 36% | 9% | 33 |
| Expanded Neutral Limits | 11 | 105 | 0.14 | 0.02 | 143 | 265 | 0.54 | 55% | 30% | 14% | 83 |
| Reduced Neutral Limits | 12 | 94 | 0.16 | 0.03 | 136 | 265 | 0.51 | 55% | 30% | 14% | 83 |
| Stability by Lapse Rate | 13 | 93 | 0.14 | 0.05 | 129 | 265 | 0.49 | 55% | 33% | 12% | 83 |
| Stability by P-G. Class. | 14 | 140 | 0.24 | 0.02 | 203 | 265 | 0.77 | 48% | 33% | 19% | 83 |
| Incl. $x_e > 50$km | 15 | 126 | -0.01 | 0.00 | 123 | 306 | 0.40 | 63% | 24% | 13% | 121 |
| Scaled to Max. Dist. | 16 | 107 | 0.14 | 0.02 | 145 | 265 | 0.55 | 55% | 30% | 14% | 83 |
| No limit of −5K/km | 17 | 109 | 0.16 | 0.02 | 151 | 265 | 0.57 | 53% | 31% | 16% | 83 |
| Eqns 4b and 5b (no min) | 18 | 1416 | -1.25 | 0.00 | 1085 | 265 | 4.10 | 54% | 23% | 23% | 83 |
| Alternate Neutral Eq. 16 | 19 | 4422 | -4.26 | 0.00 | 3293 | 265 | 12.44 | 51% | 23% | 27% | 83 |
| Momentum (Eq 17 & 18) | 20 | 114 | 0.17 | 0.02 | 159 | 265 | 0.60 | 54% | 30% | 16% | 83 |
| Momentum (Eq 20) | 21 | 227 | 0.40 | 0.02 | 333 | 265 | 1.26 | 48% | 17% | 35% | 83 |

Only the calculated to observed plume matches that originate from Syncrude1 (case 5) demonstrate good agreement between the Briggs equations and the observations (with an average ratio of 1.0 and more than half the calculated plume rise values with a factor of 2 of the observed plume rise values. This stack is the largest of the 8 stacks ($h_s = 183$ m, $d_s = 7.9$ m) and also has the highest average effluent exit velocity ($w_s = 12.0$ m s$^{-1}$). This suggests that the Briggs parameterization (as used in the GEM-MACH model) demonstrates better prediction with relatively larger stacks (>180 m) with higher volume flow rates (>500 m$^3$ s$^{-1}$). Based on 2010 inventory values, this stack emits 10 times more SO$_2$ than any of the other reported stacks. The resulting higher downwind concentrations would likely make observed plume much easier to

location and identify accurately.  For this Syncrude1 stack, the correlation coefficient and slope
of the best fit for the 17 stack-plume matches are not significantly different from zero.  Hence,
while the overall average plume rise for this stack appears accurate, the equations do not predict
individual cases of plume rise well.

**4.4.2 Stability**

Three types of tests were done to determine the effect of atmospheric stability classification on
the calculated plume rise: separation by stability class (cases 9 and 10), testing of sensitivity to
the limits of neutral classification (cases 11 and 12), and testing of other stability classification
methods (cases 13 and 14).  These tests are described in more detail below.

We first compare the calculated to observed plume rise values which occur during neutral
conditions only (case 9) and stable conditions only (case 10), with stability is based on Obukhov
length.  For the times when plumes were observed (and matched to stack sources), there were no
unstable classifications using the AMS03 tower site data (based on Obukhov length).   There are
50 stack-plume matches during neutral conditions and 33 stack-plume matches during stable
conditions.  There is no significant difference between the stack-plume comparisons for the
plume rise under neutral conditions versus stable conditions.  The ratio of average predicted
plume rise to observed plume rise is similar in both cases (0.55 compared to 0.53), and the
fraction of plume rise values less than one-half the observed values is near 55% in both cases.
Hence, the underestimation of plume rise does not seem to be dependent on predicted stability
classification.

Secondly, the sensitivity of the results to the limits of neutral conditions ($-4 < h_s/L < 0.5$) is
tested by doubling the limit values (case 11: $-8 < h_s/L < 1.0$) and halving the values (case 12:
$-2 < h_s/L < 0.25$).  The results demonstrate that the calculated plume rise values are not
strongly dependent on the choice of limits.  Doubling the limits does not change the statistics
relative to the base case, as it results in no changes in stability classification.  Halving the limits
results in a slightly lower average calculated plume rise value (136 m compared to the 143 m
base case) due to the reclassification of 5 stack-plume matches from neutral to unstable.

Finally, the results discussed in Section 4.1 suggest that there is poor agreement between the
various methods used to classify stability.  As discussed previously, the estimation of Obukhov
length based on the bulk Richardson number may be considered less accurate than an estimation
based on heat flux measurements.  We recalculate the plume rise values using the stability
classification based on the comparison of the negative temperature gradient, $-dT/dz$, to lapse
rate, $\Gamma$, (case 13) and again using the Pasquill-Gifford stability classification based on cloud
observations and wind speed (case 14).  The use of the lapse rate classification results in a
designation of predominantly stable conditions (Table 4).  This results in a small change in
average calculated plume height and a similar distribution of plume rise values compared to the
base case (with stability conditions based on the stability parameter, $h_s/L$).  Use of the Pasquill-
Gifford stability classification results in a mix of either neutral or unstable conditions.  This
reclassification of atmospheric stability results in a better agreement between calculated and
observed plume rise values, with an average ratio of 0.77.  However, nearly half (48%) of the

calculated plume rise values are below 50% of the observed values, suggesting there is still significant underestimation of plume rise, even with this reclassification of atmospheric stability.

### 4.4.3 Plume Rise Calculation Modifications

A number of modifications were made to test the sensitivity of the results to various assumptions and equations used to calculate plume rise in the base case. These include the assumption of validity of the equations beyond a given downwind distance (case 15), the estimation of maximum plume height for plumes may still be ascending at the measurement location (case 16), the effect of limits and minima used in the equations (cases 17,18), and finally an alternate plume 820 rise equation used for neutral conditions (case 19).

Firstly, as discussed above, the distance between the stack and the horizontal point of measurement of plume height is limited in this analysis to less than 50 km. Removal of this criteria (case 15) adds a further 38 stack-plume matches to the original 83 stack-plume matches in the base case. The observed plume rise values of these distant plumes are generally higher, 825 and the predicted plume rise values are lower. The resulting average ratio of calculated to observed is 0.40 (compared to 0.54 for the base case which only includes plumes that have travelled less than 50 km before measurement).

As discussed in Section 4.3, the calculated distance to maximum plume rise is less than the distance between the stack and the measurement location for all stack-plume matches. However, 830 when the distance to maximum plume rise is modified by a factor equal to the ratio of observed plume rise to calculated plume rise, approximately 13% of the plumes should reach maximum plume height further from the stack than the measurement location. To test whether this is causing an under-prediction of plume rise, we adjust the calculated plume rise values for those plumes with $x'_e > x_d$ by the ratio of adjusted distance to maximum plume rise to stack-to-835 measurement distance ($h'_B = h_B \, x'_e / x_d$). This is shown in Table 6 as case 16. The difference in statistics between this case and the base case is negligible, suggesting that the under-prediction of plume rise is not due to the observation of plumes which are still ascending.

The $-5$ K/km minimum value of $dT/dz$ used to calculate $s$ (Eq. 2) could potentially limit the plume rise. Steeper negative temperature gradients result in a smaller value of $s$, which would 840 result in higher plume rise under stable conditions. This condition is removed (case 17) and the resulting statistics are compared in Table 6. This results in a slightly higher predicted plume rise, with an average ratio of 0.57 (compared to 0.54 for the base case). Hence these results do not appear to be sensitive to this minimum value.

As discussed in Section 2.1, the minimum criteria of Eqns. 4 and 5, which are used in the GEM-845 MACH model are not used in other plume rise models, such as SMOKE. To investigate the difference between these two approaches, the plume rise is recalculated (case 18) using only the second (rightmost) term within the minimum functions of Eqns. 4 and 5. The resulting statistics are listed in Table 3. The removal of the minimum function results in 3 cases of extremely (i.e. unrealistically) high plume rise (between 6 and 41 km), all of which occur in neutral conditions. 850 Because of these extreme values, the ratio of average predicted to average observed plume rise is

4.1. However, the majority of predicted values (54%) are less than half of the observed plume rise values (similar to the base case), suggesting that the high ratio of predicted to observed value is due to a few outliers. This implies that a lower limit on wind speed and friction velocity should be used to prevent unrealistically high plume rise values when using these equations without the minimum functions, making the GEM-MACH choice of minima appropriate.

In order to test other parameterizations of plume rise, the equation for plume rise in neutral conditions (Eq. 3) is replaced by an alternative equation (De Visscher, 2013), given as

$$\Delta h = \frac{400 F_B}{U^3}.$$ (16)

The alternative equation is tested as case 19. For cases with moderately low wind speeds ($2 < U < 3$ m s$^{-1}$), the equation gives plume rise as high as 6 km, while for very low wind speeds ($U < 1$ m s$^{-1}$), plume rise higher than 100 km is predicted. This suggests this equation should be limited to cases of neutral conditions with high wind speeds, and it may be better suited for stability classification using the Pasqill-Gifford scale, which requires higher wind speeds for neutral stability classification (for non-overcast conditions).

### 4.4.4 Effluent Momentum

The plume rise due to momentum of stack effluent is not included in the parameterization used in GEM-MACH (see Section 2.1). To investigate whether neglect of momentum rise may be a significant contribution to the underestimation of plume rise we test two sets of equations to include this effect. Plumes are typically classified as either momentum driven or buoyancy driven, and the maximum of $\Delta h$ and $\Delta h_m$ is used to estimate plume rise (e.g. Briggs, 1984; VDI, 1985). As a first test, we add $\Delta h$ and $\Delta h_m$ together to give an upper limit of plume rise due to both momentum and buoyancy. As a second test, we use a parameterization (De Visscher, 2013) that includes both effects simultaneously.

For the first test (case 20), parameterizations for momentum-dominated plumes developed by Briggs are given in De Visscher (2013) for stable and neutral conditions respectively as

$$\Delta h_m = 1.5 \left( \frac{F_m}{U S^{1/2}} \right)^{1/3}, \quad \Delta h_m = 3 \left( \frac{F_m}{U^2} \right)^{1/2},$$ (17,18)

where the momentum flux is

$$F_m = \left( \frac{T_a}{T_s} \right) \frac{d_s^2 w_s^2}{4}.$$ (19)

A parameterization of the plume rise due to momentum during unstable conditions is not required here as there are no cases of plume matching during unstable conditions using the AMS03 tower data used for this comparison. Eqns. 17 and 18 are meant for plume rise due to momentum only (without buoyancy). Here we add the plume rise due to momentum to the plume rise due to buoyancy as $h_B = \Delta h + \Delta h_m$. This results in a slight improvement in predicted plume rise (ratio of 0.60 compared to the base case of 0.54), but the majority (54%) of predicted plume rise values are less than half the observed values.

For the second test (case 21) we follow the approach used in the CALPUFF model in which buoyancy and momentum are considered simultaneously (De Visscher, 2013). For plume rise in neutral or stable conditions, the plume rise can be calculated as

$$\Delta h = \left( \frac{3F_m x_e}{\beta^2 U^2} + \frac{8.3 F_b x_e^2}{U^3} \right)^{1/3} \tag{20}$$

where $x_e$ is given by Eq. 15 and $\beta = 1/3 + U/w_s$. The CALPUFF model limits the wind speed at stack height ($U$) used in Eq. 20 to a minimum of 1 m s$^{-1}$. Including this limit in our analysis had negligible effect on the resulting plume rise values. Statistics for this analysis are shown in Table 6 as case 21. The ratio of average predicted to observed values (1.26) suggests an overestimation of plume rise with this method. Nearly half (48%) of the predicted plume rise values are less than half the observed values and a large fraction (35%) of the predicted plume rise values are more than double the observed values. Hence this method seems to both overestimate and underestimate a large fraction of plume rise values, but the average predicted plume rise is closer to the average observed predicted plume rise compared to the GEM-MACH parameterization of buoyancy only.

The high fraction of under-predicted plume rise (48%) and under-predicted plume rise (35%) using the combined buoyancy/momentum formula of Eq. 20 warrants extra investigation. Of the 83 plume to stack matches used in this analysis, 40 are under-predicted (ratio < 0.5) and 29 are over-predicted (ratio > 2). Of the 40 which are under-predicted, 34 are Suncor stacks. Of the 29 that are over-predicted, 22 are Syncrude stacks. All 4 plume-to-stack matches with CNRL stacks are under-predicted. Hence there is a very strong correlation with stack location. This is consistent with the results discussed in Section 4.4.1, since the Syncrude stacks have high effluent exit velocities (e.g. Table 1), the Suncor stacks have low to moderate effluent exit velocities, and the CNRL stacks have moderate exit velocities. Combining the buoyancy and momentum with Eq. 20 appears to overestimate the influence of momentum, while simultaneously underestimating the influence of buoyancy.

## 4.5 The influence of stack-location-specific meteorological data – Companion Paper

Our focus within this work was the use of the available measurement data as a proxy for the meteorological conditions at the stack locations themselves. However, significant differences could be seen in the data between the different measurement platform locations (see Table 2). In subsequent work in our companion paper (Akingunola *et al.* 2018, this issue), high resolution meteorological model forecast simulations for the region were carried out. These suggested the presence of significant spatial heterogeneity in the meteorological parameters used to drive both the Briggs parameterization and the layered method. Predicted meteorological parameters at the meteorological measurement platform locations were substantially different from those at stack locations. When tested using the model-predicted at-stack meteorological values, and NPRI stack emissions data, the Briggs parameterization and the layered approach resulted in very different plume rise behaviour. Predicted surface $SO_2$ concentration performance was substantially improved across all metrics when the layered approach was used, and aircraft $SO_2$ comparisons improved for all metrics aside from bias. For the predicted plume heights, the slope

of the model observation line was -0.16 for the Briggs parameterization, and 0.97 for the layered approach, with the former under-predicting, and the latter over-predicting the aircraft-observation-estimated plume height. The reader is directed to Akingunola *et al*. (2018) for a discussion of these issues, which suggests that accuracy of estimates of the driving meteorological parameters at the stack locations has a controlling influence on the performance of the layered approach, and with the layered approach recommended for future development.

## 5. Conclusions

These results demonstrate a significant underestimation of plume rise using the Briggs plume rise parameterizations. The ratio of average modelled plume rise to average measured plume rise $(\overline{h_B}/\overline{h_M})$ varies from 0.55 to 0.82 using Briggs parameterization with the tower or RASS used to measure input variables. The ratio $\overline{h_B}/\overline{h_M} = 0.47$ or 0.49 using the layered method with either the RASS of the aircraft used to measure input variables. This range of ratios suggests an average underestimation between 18 and 53%. Results are improved slightly when atmospheric stability is classified using the Pasquill-Gifford system, which improves the ratio from 0.55 using dthe AMS03 tower with stability classified according to stability parameter $(h_s/L)$ to 0.77 using the Pasquill-Gifford system. Results are also improved by including plume rise due to momentum at the stack exhaust (Eq. 20), although this results in some over-prediction of plume rise, with an average ratio of $\overline{h_B}/\overline{h_M} = 1.26$ using the AMS03 tower data.

These results are in direct contrast to the many studies summarized in VDI (1985), which consistently suggest that plume rise is overestimated by the Briggs equations. The more recent study of Webster and Thomas (2002) might possibly imply an underestimation of plume rise, owing to an overestimation of surface concentration measurements using a plume rise model; however there may be other reasons for this overestimation unrelated to plume rise. The authors of the VDI report suggest that the Briggs parameterization should be reduced by a factor of 30% in neutral conditions in order to better match observations. In contrast to this suggestion, our results would be improved significantly by increasing the Briggs parameterization by a factor of 30%.

Much of the underestimation in this study appears to be driven by two stacks (Suncor 1, 3) which have relatively low effluent exit velocities. Based on a 2010 CEMA inventory, these stacks are among the list of significant $SO_2$ emitters (0.14 and 0.19 kg s$^{-1}$), although since these are yearly average inventory values, there is a possibility that the stacks were not emitting significant $SO_2$ during this specific study period. Although there is also the possibility that the plumes from these stacks are below the lowest aircraft measurement height of 150 m (and hence not observed), given the stack heights of 107 and 137 m this seems unlikely.

By far, the best results of the Briggs parametrization (as used in the GEM-MACH model) are for the largest, Syncrude1 stack. This stack emits between 11 and 40 times more $SO_2$ (2.2 kg s$^{-1}$) than the other stacks. Although the Briggs parameterization performs poorly for the smaller and moderately sized stacks, it performs well for the large stack responsible for approximately ¾ of the total emissions. Hence, any air quality assessments using the Briggs parametrization in this

region should be reasonably accurate and future improvements to the algorithms should focus on the relatively smaller stacks.

For both the Briggs parameterization and layered method and for all the measurement platforms used in this study, the correlation of parameterized plume rise to measured plume rise is low ($r^2 \leq 0.2$) and the slopes of the least-squares fits are generally less than 0.5. Carson and Moses (1969) stated that "no plume rise equation can be expected to accurately predict short term plume rise" and that their parameterizations were "to be used for general design considerations." This

statement appears to remain true nearly 50 years later and the wide use of these same equations in air quality models indicates that little improvement has been made.

The aircraft-based measurements used for this study provide only a "snapshot" of plume rise and atmospheric conditions as measurements are made on a timescale of a few hours in the morning or afternoon over the course of a few weeks in summer. However, this consistent

underestimation of plume height for these observations suggest that further investigation is warranted. Given the advancements in atmospheric measurement technology in recent decades (e.g. automated lidar, RASS, image analysis), there is an opportunity to make long-term measurements of plume rise and atmospheric conditions in an effort to improve predictability. Although the Briggs algorithms have been in use for nearly 4 decades, are used in many air-

quality models (e.g. GEM-MACH, AEROPOL, SCREEN3, CALGRID, RADM, SMOKE, and SMOKE-EU), and are widely referenced in air quality and dispersion texts (Beychok, 2005; Arya, 1998), the verification of these algorithms relies on decades old measurement techniques. More in-situ measurements of plume height are clearly needed to attempt to quantify the uncertainties in these parameterizations and to suggest improvements to the algorithm.

Further, the observations compared here demonstrated the presence of considerable horizontal heterogeneity in meteorological conditions across this region, with towers within a 10km distance providing substantially different statistics of stability conditions during the study period. This suggested that meteorological observations in close proximity to the stacks are necessary to further improve the algorithms. We examine the potential impact of this heterogeneity in our

companion paper (Akingunola *et al.*, 2018) using a high resolution meteorological model.

**Acknowledgements**

The authors wish to thank the Wood Buffalo Environmental Association (WBEA) for the use of the Lower Camp Met Tower (AMS03) and Mannix Tower (AMS05) data. The Continuing

Emission Monitoring System (CEMS) data were provided by Marilyn Albert, Ewa Przybylo-Komar, Katelyn Mackay, and Tara-Lynn Carmody of Data Management and Stewardship, Corporate Services Division, Alberta Environment and Parks. Funding for the aircraft measurement study was provided by Environment and Climate Change Canada and the Oil Sands Monitoring Program

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
