# Peer review of "A Comparison of Plume Rise Algorithms to Stack Plume Measurements in the Athabasca Oil Sands"

_Atmospheric Chemistry and Physics, 2017_

## Referee Comment (RC1) · F. DiGiovanni (Referee) · 5 Jan 2018

A review of "A Comparison of Plume Rise Algorithms to Stack Plume Measurements in the Athabasca Oil Sands" by Mark Gordon1, Paul A. Makar2, Ralf M. Staebler3, Junhua Zhang2, Ayodeji Akingunola2, Wanmin Gong2, Shao-Meng Li3

Discussion started: 20 December 2017

General Comments

Authors find that the Briggs formulations underestimates plume rise for the sources investigated.

Useful alert to potential inaccuracies in the Briggs plume rise formulae which is also used in local scale regulatory air pollution models (e.g., AERMOD).  I would recommend this for publication.

If further investigation confirms this result the implication is that this will cause dispersion models using these formulations to over estimate ground level concentrations for sources with tall stacks in flat terrain, for example.

Specific Comments

1.  In section 2.1 - Authors should provide more explanation on the limits of $L$ used to define stability classes.  For example, why the specific lower limits ($-0.25h_s$) applied to $L$ to define unstable conditions?

2.  In section 2.3 – Authors should provide more comment/investigation of uncertainties introduced by assuming NPRI inventory values for effluent velocity and exit temperature for the flaring stack.

3.  In section 2.3 – Authors should make it clear that stack characteristic values shown in Table 1 are for information purposes only and not used in the actual calculations (presumably hourly stack data was used [emission rates, exit temperatures, exit flow velocities]).

4.  In section 2.7 – the flights (e.g., box flights) seem to take up to 2 hours and so some measurements during the flight will be time-displaced from other measurements during the flight.  Although this is expected, the authors should at least comment that results, such as shown in figure 3(a) actually represent different time periods and that significant evolution of the plume could have occurred during the flight.  Are there correction methods for this?  How does this affect the measurements?

5.  In section 2.7, paragraph starting at line #441 – Authors explain attempts to match calculated plumes to observed; was this matching substantiated by say, video-recording of the plume event?  Also final sentence of that paragraph states assumption of multiple plumes having merged; was this actually observed in the field as happening?  If so, I'd suggest it be noted.

6. In section 4.1, paragraph starting at line 570 – Authors observe a significant difference between Obukhov length-based stability and temperature lapse rate-based stability. It would be useful to have comment on which is the better method to use.

7. In section 4.2, 1[st] paragraph – Authors indicate that they change input variables by an "arbitrary fraction" but it would be better to change them to the reasonable limits of their range; this would then provide a more useful ranking of model sensitivity. Some of the variations they have used do, in fact, seem like reasonable limits so are they truly arbitrary?

8. In section 5, second paragraph – Authors quote the Webster and Thomas (2002) study implies an underestimation of plume rise based (seemingly) only on an overestimation of surface concentrations. I have not reviewed that study but there can be multiple reasons that a model overestimates ground-level concentrations (for example, overly conservative emission rates). It would be useful for the authors to provide comment on how they discerned that it was only the (presumed) plume rise underestimation that led to those results.

Technical Corrections

1. Some references made in text that are not provided in the reference list:

   Holmes 2006 is the Holmes and Morawska ref?

   Bringfelt (1968) missing in ref list

   Rittmann (1982) missing in ref list

   England et al (1976) missing in ref list

   Hamilton (1967) missing in ref list

   Moore (1974) missing in ref list

2. Some references in ref list seemingly not used:

   Carson and Moores (1969) – or is this the Moses and Carson ref in the text?

   Beychok (2005)

   Makar et al refs

   Menard et al (2014)

   Pregger and Friedrich (2009)

---

## Referee Comment (RC2) · A. De Visscher (Referee) · 24 Jan 2018

General comments

Plume rise calculations are an important aspect of air dispersion modeling. Without reliable plume rise calculations, the ambient concentrations predicted by air dispersion models will not be reliable either. This paper presents uses of some of the most commonly used plume rise calculations, known as the Briggs parameterization, and tests them against plume rise values based on aircraft measurements. They find that the Briggs parameterization systematically underestimates the plume rise, and hence overestimates the ambient concentration. The conclusions are generally supported by the results, but the authors could do more to fine tune their conclusions, and provide

some context by testing some alternatives to Briggs (see next section). In particular, it would be useful to find out which of the Briggs equations is responsible for the lack of agreement. It is unfortunate that the authors chose an area with such complex geography for their validation. A test in an area with simpler geography would have been able to use more reliable meteorology. As it stands, the lack of agreement between the measured data and the predictions is at least partly due to the lack of representative meteorological parameters due to the complex terrain. The paper is clear and well-written. Overall, this is a useful contribution, and can potentially be a very good paper if a number of modifications are made. Detailed comments are given in the next sections.

Specific comments

Note: Many comments here are essentially criticisms of the Briggs parameterization. These are not meant as criticisms directed at the authors.

- p. 4 lines 133-134: The authors used the temperature gradient between the surface and the stack tip as a proxy for the temperature gradient above the stack tip. This will cause the atmospheric stability to appear less neutral than it actually is (i.e., more stable, as s is meant to be used in stable atmosphere).

- p. 4 line 135: when the "maximum" temperature gradient is set at -5 K/km, do you mean -5 is the least negative gradient (i.e., the most stable gradient)? Please provide a reason for this choice, as more stable atmospheres are quite common. Also, it would be useful to test the effect of this restriction on the plume rise predictions.

- eq. 4 p. 4: I realize that most air dispersion models define a final plume rise for unstable atmosphere, but I find this a fundamentally flawed notion: in an unstable atmosphere the plume will continue rising until it either approaches the top of the mixing layer, or gets trapped in a downdraft stronger than the plume's rise. Given this, it is not surprising that the Briggs parameterization tends to underestimate plume rise. I would argue that the parameterization was designed to underestimate plume rise.

Given this, it is surprising that earlier studies indicated that the Briggs parameterization overestimated plume rise. It would be useful to gain insight as to why earlier studies found plume rises less than predicted by Briggs. Were these also final plume rise calculations, or transitional plume rise?

- eq. 4 seems to predict unrealistically small plume rise when the wind speed is high. Also, depending on what friction velocities are used in unstable vs neutral plume rise, the neutral plume rise equation (eq. 5) often predicts larger plume rise than the unstable plume rise equation. That seems unrealistic to me.

- Eq. 5 p. 5 contains an error. $u^*$ in denominator of the last term should be squared. Please check that the calculations were carried out correctly.

- Eqs. 4 p. 4 and 5 p. 5: the second function of eq. 4 and the first function of eq. 5 are dimensionally not homogeneous, which means they are not supported by similarity considerations, and they will not have a broad validity. Please bear this in mind.

- For both eq. 4 and eq. 5, the second function of the minimum seems more realistic to me. It would be useful to check if these second functions provide better predictions than the first functions.

- An equation that is sometimes used for final plume rise in a neutral atmosphere is $400 \, Fb/U^3$. It might be useful to check if that equation gives any better predictions than eq. 5.

- eq. 7 only makes sense in a stable atmosphere, because s only has physical meaning in a stable atmosphere. Was it used for stable atmosphere only, or for all types of atmosphere?

- p. 6 lines 208-209: For some emissions, it was assumed that the emission profile in 2013 was the same as in 2010. That puts the calculation on shaky ground. Do you really need these data?

- Table 1 p. 7: Please indicate which data were collected with CEMS, and which

weren't. The CEMS data will be a lot more reliable, and should be treated as such.

- Table 1 p. 7: If I understood correctly, you mention 19 emissions here, but you only used 8 of them. Unless I misunderstood or unless you have a compelling reason to keep all the data in the table, please remove the data that were not used in the test.

- eq. 8 p. 7: Why not use an equation based on the momentum flux parameter Fm for plume rise due to momentum? (see p. 22)

- Table 2: Some correlations seem quite low. To what extent is the lack of agreement between the predictions and the measured plume heights due to wind and temperature uncertainties?

- eq. 9 p. 10: Please check if this is correct. Richardson numbers are normally based on the potential temperature, not the actual temperature. Please also check the other variables in the equation and make sure they were interpreted correctly.

- eq. 10 p. 10: what value of z is used here?

- eq. 10 p. 10: Estimating L without a sensible heat flux measurement or estimation is very difficult. Expect substantial inaccuracies with this equation. This may explain why the values of z0 vary so strongly by location. A value of 10.1 m, for instance, is suspiciously high even for a forest.

- eq. 11 p. 10: Also expect substantial inaccuracies for this equation. At the verge of a temperature inversion, this equation predicts infinite boundary-layer height. In an unstable atmosphere, the boundary layer height is mainly influenced by the accumulated sensible heat deposited into the atmosphere during the current day, so parameterizations such as eq. 11 are questionable in unstable atmospheres.

- Figure 4 shows a distribution of the calculated plume rise values, for the different calculation schemes and input data. How do these distributions compare with measured plume rise values?

- p. 16: Comparing the average ratio between predictions and measurements of the plume rise will tend to be biased, because a small number of data points with very low measured plume rise (small denominator) can skew the results upwards. To complement this information, it would be best to also calculate the average calculated plume rise, and compare it with the average measured plume rise. This will tend to give the instances of high plume rise the largest weight, so it is also an imperfect measure. Reporting both average ratio and ratio of the averages will give the reader a good sense of how the measurements compare with the calculations.

- Figure 5: It could be coincidence, but I have the impression that there is some clustering of the data points, particularly near the x axis (very low predictions irrespective of the actual plume rise). This gives me the impression that some equations within the Briggs parameterization are far less accurate than others. It would be useful to see the performance of each equation separately (even distinguishing between the two equations where a minimum is calculated). Also, if some of these data are based on CEMS and some are based on emission inventory data, it would be useful to know which is which, because the CEMS data will be much more reliable. I realize that I'm asking for a lot of disaggregation here. Perhaps a supplementary document could be prepared alongside the paper.

- Table 4 p. 19: there is a huge discrepancy between the stabilities evaluated from the data of the different sources. This confirms the poor reliability of the calculation scheme for L. If the Pasquill stability classes are known for these measurements, then it might be possible to determine which data set is most reliable.

- Table 5 p. 20: This sensitivity analysis is very useful, but I find the result suspect. The surface temperature is found to have almost no influence on the plume rise, but the value of H has a large effect. The surface temperature affects H quite strongly, so I don't see how this is possible. Please check.

- p. 22, top: The authors claim that final plume rise is reached within 2 km in all cases.

I find that hard to believe when some plumes rise by 600-800 m. If the Briggs parameterization greatly underestimates plume rise, it will also underestimate the distance to final plume rise. Hence, I would suggest that the authors use the maximum measured plume rise as a guide for estimating the maximum distance to plume rise.

- p. 22 line 668: Please capitalize letter D in my name and sort my book under D in the reference list, not under V. Eq. 17 on line 670 is useful, but I suggest checking out eq. (15.69) on p. 533 of my book as well (after correcting the typos: the factor $x^2$ should not be there, and the factor us in both denominators should not be squared). This equation, as used by CALPUFF, gives predictions of final plume rise when both momentum and buoyancy affect plume rise.

Technical corrections

- p. 6 line 184: delete "from"

- Figure 1 (b) p. 8: the scale on this figure is off by about a factor 2. The scale on Figure 1 (a) seems OK.

- Table 2 caption states that AMS03 measurement height is 90 m, whereas the text (line 286) states it's 167 m.

- p. 23 line 718: Beychok and Milton, 2005 should read Beychok, 2005.

- Please check reference lists of other papers for the correct abbreviations of journal titles. For instance, Atm. Env. should read Atmos. Environ.
* * *

---

## Author Response (AR1)

General Comments from the Authors:

We would like to sincerely thank both reviewers for their instructive comments and we believe the manuscript has been significantly strengthened based on these comments and ideas. We have tried in every instance to incorporate these comments into a revised analysis and manuscript. Responses to each suggestion or questions are given below.

There were two significant errors in the previously submitted manuscript. One of which (pointed out by Dr. De Visscher) is the use of temperature in the Obukhov length and boundary-layer height calculations, where the actual equations specify potential temperature. The second error was the comparisons of Fig. 5 and Table 3 were actually comparisons of effective plume height and not plume rise as stated (i.e. the numbers included stack heights and were hence offset between 76 and 183 m. These errors have been corrected our revised manuscript. Although the numbers have changed due to these corrections, the general findings of the study (that the Briggs equations significantly underestimate the plume rise at this location) is still supported by these new results.

Further, we have expanded on one of Dr. De Visscher's suggestions to disaggregate the results and we used this new analysis to address a number of other reviewer comments and concerns. The revised manuscript now contains a new section in which the comparison of calculated to observed plume rise is redone 24 times, under a variety of separation of data (i.e. comparisons for each stack and for each stability class) as well as variations on the original equations (i.e. alternate plume rise formulae, removal of the minimum functions, different ways to calculate stability, and more). The statistical summary of these comparisons will be presented as a new table in the revised manuscript (following the style of Table 3) and the comparison plots (as Fig 5.) for all 24 cases will be provided as supplementary material. In the comments below, this reanalysis is referred to repeatedly as the "conditions comparison".

Responses to individual reviewer comments follow in blue text.

RC1:

1. In section 2.1 - Authors should provide more explanation on the limits of L used to define stability classes. For example, why the specific lower limits (-0.25hs) applied to L to define unstable conditions?

These limits of $z/L$ are given by Briggs and a citation is added to the text. It is noted that Briggs suggests using the effective stack height ($h_e = h_s + \Delta h$), which would be difficult to code (since stability class becomes dependent on plume rise, which is dependent on stability class). GEM-MACH uses $z = h_s$ instead. In the new "conditions comparison" (see above) we explore the effect of changing the limits of neutral conditions (to both $-8 < \frac{h_s}{L} < 1$ and $-2 < \frac{h_s}{L} < 0.25$). Both these changes result in minimal change in the resulting predicted plume rise, suggesting the analysis is not sensitive to the choice of these parameters.

2. In section 2.3 – Authors should provide more comment/investigation of uncertainties introduced by assuming NPRI inventory values for effluent velocity and exit temperature for the flaring stack.

We have added a comparison using the other CNRL stack (CNRL1), which is a sulphur recovery unit. For CNRL1, both CEMS and NPRI data are available for this period, allowing a direct comparison between the two reporting methods. This comparison demonstrates that the reported NPRI temperature is within 5% of the CEMS temperature, the NPRI reported exit velocity is more than a factor of 4 higher than the CEMS exit velocity. While the flaring stack (CNRL4 in the previous manuscript) may be very different than the sulphur recovery unit stack, this at least provides a rough estimate of the uncertainty due the use of NPRI values.

3. In section 2.3 – Authors should make it clear that stack characteristic values shown in Table 1 are for information purposes only and not used in the actual calculations (presumably hourly stack data was used [emission rates, exit temperatures, exit flow velocities]).

New text has been added to revised manuscript to clarify – both within the main text (2nd paragraph) and the table caption.

4. In section 2.7 – the flights (e.g., box flights) seem to take up to 2 hours and so some measurements during the flight will be time-displaced from other measurements during the flight. Although this is expected, the authors should at least comment that results, such as shown in figure 3(a) actually represent different time periods and that significant evolution of the plume could have occurred during the flight. Are there correction methods for this? How does this affect the measurements?

A paragraph is added (6th paragraph) to discuss the effects of stationarity. Reference is made to Gordon et al. (2015), where the effects are discussed in more detail. It is expected that this would have little effect on the estimated location of the plume centre, especially as the direction of spiraling was not consistent between flights (i.e. the aircraft would sometimes fly in an ascending spiral and sometimes in a descending spiral).

5. In section 2.7, paragraph starting at line #441 – Authors explain attempts to match calculated plumes to observed; was this matching substantiated by say, video-recording of the plume event? Also final sentence of that paragraph states assumption of multiple plumes having merged; was this actually observed in the field as happening? If so, I'd suggest it be noted.

There was no video recording of plume events. Matching is done purely through proximity of observed and parameterized coordinates. Text is added to note that visual observations by the authors during the field work supports the plume merging, especially far downwind of the stacks.

6. In section 4.1, paragraph starting at line 570 – Authors observe a significant difference between Obukhov length-based stability and temperature lapse rate-based stability. It would be useful to have comment on which is the better method to use.

The criteria for stability are now tested in the "conditions comparison" with both Obukhov length and lapse rate (see Table 4). Based on the second reviewer's comments we have also added a comparison of the Pasquil-Gifford stability class. All three methods produce very difference distributions of stability class, but all methods result in a significant underprediction of plume rise by the Briggs equations. Hence the main conclusions of the study are not strongly dependent on the stability calculation used.

7. In section 4.2, 1st paragraph – Authors indicate that they change input variables by an "arbitrary fraction" but it would be better to change them to the reasonable limits of their range; this would then provide a more useful ranking of model sensitivity. Some of the variations they have used do, in fact, seem like reasonable limits so are they truly arbitrary?

This section has been redone so that the variations are calculated based on the difference between the AMS03 and AMS05 towers. The plume rise is then recalculated for the duration of the study modified by this average difference. This demonstrates the sensitivity of the plume rise calculation to the heterogeneity of the variables within the study area. As would be expected there are large differences in the values of $H$ and $L$ calculated with the AMS03 data relative to the values of $H$ and $L$ calculated with the AMS05 data (71% and 165%). There are significant differences in the average plume rise (between –27% and 7%) when the $H$ and $L$ values data are modified by these amounts. We believe this makes the modification less arbitrary, as the data are being modified by an actual difference in measured values, as opposed to a "hand-picked" percentage. We also note that the issue of spatial heterogeneity of the meteorological observations was shown to be of key importance in our companion paper (Akingunola et al), which examines the plume rise equations within the context of a high resolution coupled air-quality/meteorology model (GEM-MACH).

8. In section 5, second paragraph – Authors quote the Webster and Thomas (2002) study implies an underestimation of plume rise based (seemingly) only on an overestimation of surface concentrations. I have not reviewed that study but there can be multiple reasons that a model overestimates ground-level concentrations (for example, overly conservative emission rates). It would be useful for the authors to provide comment on how they discerned that it was only the (presumed) plume rise underestimation that led to those results.

The language used to describe the Webster and Thomas results and their interpretation is modified to emphasize that this is only one possible reason for the overestimation.  For example, lines in Section 1 now state "However, there may be other factors…" and in Section 5 "however, there may be other reasons…".

9. Various references.

Fixed.

RC2 Specific Comments:

Note: Many comments here are essentially criticisms of the Briggs parameterization. These are not meant as criticisms directed at the authors.

We thank the reviewer for this explanation. Generally, we have tried to incorporate as many of these comments as possible into the discussion, especially in light of the poor performance of the Briggs algorithms in this analysis. Whenever possible, the new "conditions comparison" (as discussed above) is used to test some aspects of the parameterization that are pointed out as potentially flawed or inaccurate.

- p. 4 lines 133-134: The authors used the temperature gradient between the surface and the stack tip as a proxy for the temperature gradient above the stack tip. This will cause the atmospheric stability to appear less neutral than it actually is (i.e., more stable, as s is meant to be used in stable atmosphere).

This is how the temperature gradient is calculated in the GEM-MACH model, so we are following this approach here. Text is added here to acknowledge this fact and to point out that the "layered method" described later in Section 2.2 is essentially a test of using known temperature gradients above the stack height.

- p. 4 line 135: when the "maximum" temperature gradient is set at -5 K/km, do you mean -5 is the least negative gradient (i.e., the most stable gradient)? Please provide a reason for this choice, as more stable atmospheres are quite common. Also, it would be useful to test the effect of this restriction on the plume rise predictions.

The –0.005 K/m gradient is in the code statement dTdz=max(dTdz,–0.005), so this value is a minimum - not a maximum as stated (we thank the reviewer for bringing this error to our attention). It is the least stable gradient that can be used to calculate $s$. This is an approximation of the moist (pseudo) adiabatic lapse rate. The statement is a requirement of both the plume rise algorithms themselves and any attempt to convert them into code, in that the use of this minimum avoids the possibility of a value of $s = 0$ as the environmental lapse rate approaches this limit. Otherwise, the algorithm would give an infinite plume rise if the Obukhov length indicated stable conditions. We have added text to help make this clear, including the resulting condition of $s \geq 0.047/T_a$. The "conditions comparison" tests the analysis without this minimum condition and it is found that the change in predicted to observed plume rise is small.

- eq. 4 p. 4: I realize that most air dispersion models define a final plume rise for unstable atmosphere, but I find this a fundamentally flawed notion: in an unstable atmosphere the plume will continue rising until it either approaches the top of the mixing layer, or gets trapped in a downdraft stronger than the plume's rise. Given this, it is not surprising that the Briggs parameterization tends to underestimate plume rise. I would argue that the parameterization was designed to underestimate plume rise. Given this, it is surprising that earlier studies indicated that the Briggs parameterization overestimated plume rise. It would be useful to gain insight as to why earlier studies found plume rises less than predicted by Briggs. Were these also final plume rise calculations, or transitional plume rise?

This is a very interesting point.  However, as noted in a previous comment, the stability class is based on measurements near the surface, without knowledge of conditions further aloft.  The details of the previous studies are not clear, but it is noted that most of the corrections are given for neutral conditions only.  Based on the shortcomings that the reviewer points out, it would be expected that the "layered method" should perform much better, since this method uses measurements of gradients throughout the mixing layer.  If the layered method were used in a fully unstable boundary layer the plume would rise to the top of the mixing layer.  This is a better representation of what the reviewer states.  But the results show that the layered method actually gives lower plume rise than the Briggs approach, implying that no completely unstable boundary layers were observed in this study.

- eq. 4 seems to predict unrealistically small plume rise when the wind speed is high. Also, depending on what friction velocities are used in unstable vs neutral plume rise, the neutral plume rise equation (eq. 5) often predicts larger plume rise than the unstable plume rise equation. That seems unrealistic to me.

We agree that the Briggs approach seems flawed, but we are testing the parameterization as it appears in the models.

- Eq. 5 p. 5 contains an error. u* in denominator of the last term should be squared. Please check that the calculations were carried out correctly.

This is a typo in the manuscript only and has been corrected.  The equation is correct in the programs used for the analysis.

- Eqs. 4 p. 4 and 5 p. 5: the second function of eq. 4 and the first function of eq. 5 are dimensionally not homogeneous, which means they are not supported by similarity considerations, and they will not have a broad validity. Please bear this in mind.

This is true and is another shortcoming of the Briggs equations (or at least the variants used in GEM-MACH).

- For both eq. 4 and eq. 5, the second function of the minimum seems more realistic to me. It would be useful to check if these second functions provide better predictions than the first functions.

This check had already been done as part of the testing described in Section 4.4.1 of the submitted manuscript.  It is easy to see how that detail might have been missed, especially as this section appears much later in the paper.  This analysis is now included as part of the "conditions comparison".  With the revised analysis, it turns out that removing the minimum functions for these data results in unrealistically high plume rises, so use of the minimum functions is recommended as part of the manuscript conclusions.

- An equation that is sometimes used for final plume rise in a neutral atmosphere is 400 Fb/U^3. It might be useful to check if that equation gives any better predictions than eq. 5.

This alternate formula is included as part of the "conditions comparison" analysis. In cases of low wind speeds, it results in extreme values of plume rise. Hence, based on these results its use is not recommended for use.

- eq. 7 only makes sense in a stable atmosphere, because s only has physical meaning in a stable atmosphere. Was it used for stable atmosphere only, or for all types of atmosphere?

Text is added to clarify that the equation is used for only stable and neutral layers ($s = 0$ in a neutral layer). This approach was explained in our companion paper in this special issue (Ayodeji et al.) and that explanation is reproduced in this section. In summary: the plume is assumed to ascend without loss of buoyancy in unstable layers (as with the neutral case). However, it is noted (as discussed in Section 4.1) that the majority of the layers are either neutral or stable (see Table 5).

- p. 6 lines 208-209: For some emissions, it was assumed that the emission profile in 2013 was the same as in 2010. That puts the calculation on shaky ground. Do you really need these data?

The 2010 inventory info was only used to determine $SO_2$ mass emission rate, not stack parameters, as a means of selecting the stacks with the greatest potential impact on $SO_2$ concentrations for further analysis. This value is either significant for a stack that emits $SO_2$, or negligible for a stack that does not. This is the criteria used to reduce the total number of stacks to 8 stacks of interest due to their level of emissions. These values are not used in any calculations – only selection. It was assumed that a stack designed to emit $SO_2$ in 2010 will still be a significant emitter of $SO_2$ in 2013. Since the table is reduced to only include the 8 stacks used in the analysis (following the reviewer's suggestion below), the text to explain this has been modified and is hopefully easier to understand.

- Table 1 p. 7: Please indicate which data were collected with CEMS, and which weren't. The CEMS data will be a lot more reliable, and should be treated as such.

Only the CNRL flare stack (originally called CNRL4, now called CNRL2) flow rate and temperature are from NPRI inventory values (i.e. not CEMS). This is labelled in the new table in a way that is easier to see. This stack is also disaggregated from the results in the "conditions comparison" section (as are all the stacks).

- Table 1 p. 7: If I understood correctly, you mention 19 emissions here, but you only used 8 of them. Unless I misunderstood or unless you have a compelling reason to keep all the data in the table, please remove the data that were not used in the test.

The original intention was to demonstrate why we only select the 8 stacks of interest (based on $SO_2$ emissions and observations). However; we agree with the reviewer that including the statistics for all these unused stacks adds no value to the manuscript, so the explanation for the selection of the 8 stacks is in the text only. The section has been rewritten to more clearly explain the rationale and process of selecting the 8 stacks for analysis.

- eq. 8 p. 7: Why not use an equation based on the momentum flux parameter Fm for plume rise due to momentum? (see p. 22)

We have moved the analysis of momentum from this section. It is discussed in the "conditions comparison" section along with some other parameterizations (including those based on the momentum flux parameter).

- Table 2: Some correlations seem quite low. To what extent is the lack of agreement between the predictions and the measured plume heights due to wind and temperature uncertainties?

We discuss the fact that this is a very inhomogeneous terrain and hypothesize that this is the reason for the low correlation. This will undoubtedly lead to lower agreement between the predicted and measured plume heights, and this may be responsible for the very low correlations between modelled and measured plume rise values (see Table 3). The sensitivity tests (Section 4.2) have been rewritten (see Reviewer 1, comment 7), so that they test what effect the difference between measurement locations can have on the plume rise. While the lack of agreement between predicted and measured plume height may be due to different measurement locations, Fig. 5 and Table 3 demonstrate that all of the measurement locations result in (nearly) the same underprediction of plume rise. In our companion paper (Akingunola et al), we have used a high resolution simulation of the on-line GEM-MACH air-quality model to examine potential impacts of the large level of horizontal variability of the temperature profiles on predicted plume rise, the model having the advantage of providing continuous profiles at the actual stack locations themselves.

- eq. 9 p. 10: Please check if this is correct. Richardson numbers are normally based on the potential temperature, not the actual temperature. Please also check the other variables in the equation and make sure they were interpreted correctly.

This was an error (as discussed in the opening paragraphs of this response). We have corrected the analysis to use potential temperature for both the Richardson number and the boundary-layer height calculations. As discussed above, the main conclusions of the manuscript do not change because of this correction. We have also double checked all the other variables and confirmed that they are correct.

- eq. 10 p. 10: what value of z is used here?

We add the text: "The Obukhov length is calculated from the stability parameter as $L = z_{max}/(z/L)$, where $z_{max}$ is the highest measurement height of 167 m, 90 m, or up to 800 m for AMS03, AMS05, and the RASS respectively.".

- eq. 10 p. 10: Estimating L without a sensible heat flux measurement or estimation is very difficult. Expect substantial inaccuracies with this equation. This may explain why the values of z0 vary so strongly by location. A value of 10.1 m, for instance, is suspiciously high even for a forest.

We agree this is not a very accurate method for the estimation of $L$. We have added a paragraph to the text to discuss this. The magnitude of $L$ is not used directly in the Briggs equations, except in the determination of stability class and convective velocity ($H_*$). The new sensitivity analysis demonstrates that $L$ varies by an average of 165% between the AMS03 and AMS05 stations. We

have added text to note that this is likely due to the uncertainties in this equation and also to suggest the potential explanation for the very high $z_o$ value.

- eq. 11 p. 10: Also expect substantial inaccuracies for this equation. At the verge of a temperature inversion, this equation predicts infinite boundary-layer height. In an unstable atmosphere, the boundary layer height is mainly influenced by the accumulated sensible heat deposited into the atmosphere during the current day, so parameterizations such as eq. 11 are questionable in unstable atmospheres.

We also agree this is not a very accurate method for the estimation of $H$ and we have added text to discuss this in the revised manuscript.

- Figure 4 shows a distribution of the calculated plume rise values, for the different calculation schemes and input data. How do these distributions compare with measured plume rise values?

We had been hesitant to do this, as this section was meant as a comparison of plume rise as calculated with data from the various measurement locations. These distributions are for the entire flight period (the 46 hours when box or screen patterns were being flown). The next section then compared the plumes which could be matched to specific stacks. To calculate plume rise from the observations it is necessary to match this plume with an emitting stack so that the stack height can be subtracted from the observed height. Hence what the reviewer asks for here is essentially a comparison of two different things (plume heights for every stack over a 46 hour period versus only matched, observed plume heights).

- p. 16: Comparing the average ratio between predictions and measurements of the plume rise will tend to be biased, because a small number of data points with very low measured plume rise (small denominator) can skew the results upwards. To complement this information, it would be best to also calculate the average calculated plume rise, and compare it with the average measured plume rise. This will tend to give the instances of high plume rise the largest weight, so it is also an imperfect measure. Reporting both average ratio and ratio of the averages will give the reader a good sense of how the measurements compare with the calculations.

Average calculated and average predicted have been added to all the relevant tables and are discussed in the text.

- Figure 5: It could be coincidence, but I have the impression that there is some clustering of the data points, particularly near the x axis (very low predictions irrespective of the actual plume rise). This gives me the impression that some equations within the Briggs parameterization are far less accurate than others. It would be useful to see the performance of each equation separately (even distinguishing between the two equations where a minimum is calculated). Also, if some of these data are based on CEMS and some are based on emission inventory data, it would be useful to know which is which, because the CEMS data will be much more reliable. I realize that I'm asking for a lot of disaggregation here. Perhaps a supplementary document could be prepared alongside the paper.

We agree that this is a lot of disaggregation; however we have attempted to incorporate as much of this reanalysis as possible in the revised manuscript. The CEMS versus NRPI analysis is only

the removal of one stack (was CNRL4, is now CNRL2). However, based on this and other comments we have opted to add the new "conditions comparison" analysis (as discussed above) in which the comparison between predicted and observed plume rise has been repeated under multiple new scenarios. As per the reviewer's suggestion, we also present the variation of Figure 5 for each of these variations as supplementary data.

With respect to the clustering noted by the reviewer, this is a results of the error discussed above in which effective plume height was plotted in this figure instead of plume rise. Having fixed this error in the revised manuscript, it is apparent that this is a clustering of very low plume rises (near 0), due to two stacks with very low exit velocity (Suncor 1 and 3 in Table 1). The results are also analyzed without these stacks as part of the "conditions comparison" and the implications are discussed in the revised manuscript.

- Table 4 p. 19: there is a huge discrepancy between the stabilities evaluated from the data of the different sources. This confirms the poor reliability of the calculation scheme for L. If the Pasquill stability classes are known for these measurements, then it might be possible to determine which data set is most reliable.

We have added estimations of the Pasquill stability class to Table 4. As a test of the effect of uncertainty in determining stability class, we have added a reanalysis of the data using lapse rate and Pasquill stability classes to determine the effect of stability classification technique on plume rise. Use of the Pasquill stability class does improved the results somewhat, but there is still an underprediction of plume rise by the Briggs equations with all three techniques (i.e. Obukhov length, lapse rate, and Pasquill class).

- Table 5 p. 20: This sensitivity analysis is very useful, but I find the result suspect. The surface temperature is found to have almost no influence on the plume rise, but the value of H has a large effect. The surface temperature affects H quite strongly, so I don't see how this is possible. Please check.

This analysis was done by changing the value of the available input parameters in the plume algorithm (equivalent to multiplying the predetermined value of $T_{surface}$ or $H$ in the input file). The values are not modified "from the beginning", so $H$ is not recalculated with the modified $T_{surface}$, for example. As the reviewer has noted, this is not exactly correct, so we have recalculated the sensitivity by modifying the variables before all calculations. However, we have changed the analysis to modify $\Delta T = T_a - T_{surface}$ by a given factor (not the surface temperature alone), This demonstrates the relative importance of finding a representative temperature gradient to drive the equations.

- p. 22, top: The authors claim that final plume rise is reached within 2 km in all cases. I find that hard to believe when some plumes rise by 600-800 m. If the Briggs parameterization greatly underestimates plume rise, it will also underestimate the distance to final plume rise. Hence, I would suggest that the authors use the maximum measured plume rise as a guide for estimating the maximum distance to plume rise.

We have added a discussion based on this idea, which we are in agreement with. We have added a calculation of distance to plume rise using the observed values of plume rise from the matching

plume.  The text is rewritten to incorporate this change and a reanalysis of the data is tested in the "conditions comparison" in which the plumes which are not predicted to reach maximum height at the measurement location are scaled to their predicted height at the measurement location.  However, this correction does not appear to affect the overall results significantly.

- p. 22 line 668: Please capitalize letter D in my name and sort my book under D in the reference list, not under V. Eq. 17 on line 670 is useful, but I suggest checking out eq. (15.69) on p. 533 of my book as well (after correcting the typos: the factor $x^2$ should not be there, and the factor us in both denominators should not be squared). This equation, as used by CALPUFF, gives predictions of final plume rise when both momentum and buoyancy affect plume rise.

The book reference is corrected and the alternate momentum equation has been added to the "conditions comparison" analysis.

Technical corrections.

All the technical corrections have been incorporated into the revised manuscript.

**A Comparison of Plume Rise Algorithms to Stack Plume Measurements in the Athabasca Oil Sands**

Mark Gordon[1], Paul A. Makar[2], Ralf M. Staebler[3], Junhua Zhang[2], Ayodeji Akingunola[2], Wanmin Gong[2], Shao-Meng Li[3]

1) Earth and Space Science and Engineering, York University
2) Air Quality Modelling and Integration Section, Air Quality Research Division, Atmospheric Science and Technology Directorate, Science and Technology Branch, Environment and Climate Change Canada.
3) Air Quality Processes Research Section, Air Quality Research Division, Atmospheric Science and Technology Directorate, Science and Technology Branch, Environment and Climate Change Canada.

**Abstract**

Plume rise parameterizations calculate the rise of pollutant plumes due to effluent buoyancy and exit momentum. Some form of these parameterizations is used by most air quality models. In this paper, the performance of the commonly used Briggs plume rise algorithm was extensively evaluated, through a comparison of the algorithm's results when driven by meteorological observations with direct observations of plume heights in the Athabasca oil sands region. The observations were carried out as part of the Canada-Alberta Joint Oil Sands Monitoring Plan in August and September of 2013. Wind and temperature data used to drive the algorithm were measured in the region of emissions from various platforms, including two meteorological towers, a radio-acoustic profiler, and a research aircraft. Other meteorological variables used to drive the algorithm include friction velocity, boundary-layer height, and the Obukhov length. Stack emissions and flow parameter information reported by Continuous Emissions Monitoring Systems (CEMS) were used to drive the plume rise algorithm. The calculated plume heights were then compared to interpolated aircraft $SO_2$ measurements, in order to evaluate the algorithm's prediction for plume rise. We demonstrate that the Briggs algorithm, when driven by ambient observations, significantly underestimated plume rise for these sources, with more than 50% of the predicted plume heights falling below half the observed values from this analysis. With the inclusion of the effects of effluent momentum, the choice of different forms of parameterizations, and the use of different stability classification systems, this essential finding remains unchanged. In all cases, approximately 50% or more of the predicted plume heights fall below half the observed values. These results are in contrast to numerous plume rise measurement studies published between 1968 and 1993. We note that the observations used to drive the algorithms imply the potential presence of significant spatial heterogeneity in meteorological conditions; we examine the potential impact of this heterogeneity in our companion paper (Akingunola et al, 2018). It is suggested that further study using long-term in-situ measurements with currently available technologies is warranted to investigate this discrepancy, and that wherever possible, driving meteorological observations are conducted in the immediate vicinity of the emitting stacks.

**Commented [MG1]:** New analysis looks at modifications to the original parameterization.

**Commented [MG2]:** There is an added focus on the effect of heterogeneity in the region, which is examined in the companion paper.

[revised manuscript text omitted]

**Commented [MG4]:** The lower boundary was given as hs (stack height) in the previous manuscript. This is corrected.

Equation 7 is intended for use with stable ($s > 0$) or neutral ($s = 0$) layers. For unstable layers
we follow the approach outlined in our companion paper (Akingunola et al., 2018) in which the
plume rises through the unstable layer without gaining or losing buoyancy or momentum
(equivalent to $s = 0$ in Eq. 7). As is discussed below (Section 4.1), the majority of layer
195    temperature profiles (>90%) measured by the aircraft were stable or neutral, so this assumption
should not have a significant effect on the resulting plume rise. However, we also hound that the
stability was spatially heterogeneous in the study region, with significant differences in stability
noted from the different sources of meteorological information.

**Commented [MG5]:** An added discussion of how the model handles unstable profiles and the effect this might have on the results.

[revised manuscript text omitted]

Commented [MG7]: A comparison to another stack is added to estimate the uncertainty in the NPRI estimated parameters.

Commented [MG8]: Stacks which do not emit significant SO2 are not used in the analysis and have been removed from Table 1.

The relatively high flow rates and diameters of some stacks may lead to plume rise due to momentum alone, especially under stable conditions.  Briggs also developed similar equations for rise due to momentum (c.f. Briggs, 1984).  These equations are typically used when $F_b = 0$, and the plume is assumed to be either a vertical jet (momentum driven) or a bent over plume (buoyancy driven).  The potential effect of momentum on the plume rise is discussed in Section 4.4.

**Commented [MG9]:** There was an brief analysis of rise due to momentum here.  This analysis is done later 
[revised manuscript text omitted]

---

## Author Response (AR2)

Response to Editor's Comments

Editor's Comments (responses in blue text):

This is very interesting study demonstrating the challenges of simulating the interaction between meteorology and plume emission conditions and should be of considerable interest to air quality modellers.

Thank you to the author's for their detailed responses and the revised manuscript, which is acceptable for publication after technical corrections.

A couple of areas that the authors may consider in their revisions:

(1) In one of the sensitivity analysis the results showed that: "Nearly half (48%) of the predicted plume rise values are less than half the observed values and a large fraction (34%) of the predicted plume rise values are more than double the observed values." - This seems like a rather large dichotomy, which could potentially be explained with available information leading to some insight. The authors should consider exploring this result and briefly sharing what may be the reason(s).

We have looked into the data in more detail and it appears the dichotomy is related to the stack sources and their different effluent exit velocities. This is an interested result that and the following text is added at line 902 (at the end of Section 4.4.4)..

"The high fraction of underpredicted plume rise (48%) and underpredicted plume rise (35%) using the combined buoyancy/momentum formula of Eq. 20 warrants extra investigation. Of the 83 plume to stack matches used in this analysis, 40 are underpredicted (ratio < 0.5) and 29 are overpredicted (ratio > 2). Of the 40 which are underpredicted, 34 are Suncor stacks. Of the 29 that are overpredicted, 22 are Syncrude stacks. All 4 plume-to-stack matches with CNRL stacks are underpredicted. Hence there is a very strong correlation with stack location. This is consistent with the results discussed in Section 4.4.1, since the Syncrude stacks have high effluent exit velocities (e.g. Table 1), the Suncor stacks have low to moderate effluent exit velocities, and the CNRL stacks have moderate exit velocities. Combining the buoyancy and momentum with Eq. 20 appears to overestimate the influence of momentum, while simultaneously underestimating the influence of buoyancy."

(2) While a strength of this study is the use of observed/measured inputs in the plume rise formulations, which tended to show an underestimate of plume rise and little skill in predicting rise for individual plumes, it could be useful to readers to also see plume rise results for these cases based upon using the modeled (GEM-MACH) inputs. Perhaps this information is in the identified companion paper, however, it would be of value to briefly present these results in this paper so that all the information is in one place. Could an additional line be included in Figure 3 and/or the statistics in Table 3? If these results are added then some text would be important to describe the result. However, ideally, further linking to the companion paper would avoid the need for too much additional revision in this paper to explain what was done.

This information does indeed appear in the companion paper. We have added a brief summary of some of the main results of that paper in a new section at the end of the Discussion at line 914. (just before the Conclusions), as well as directing the reader to that paper for the details:

"**4.5 The influence of stack-location-specific meteorological data – Companion Paper**

Our focus within this work was the use of the available measurement data as a proxy for the meteorological conditions at the stack locations themselves. However, significant differences could be seen in the data between the different measurement platform locations (see Table 2). In subsequent work in our companion paper (Akingunola *et al.* 2018, this issue), high resolution meteorological model forecast simulations for the region were carried out. These suggested the presence of significant spatial heterogeneity in the meteorological parameters used to drive both the Briggs parameterization and the layered method. Predicted meteorological parameters at the meteorological measurement platform locations were substantially different from those at stack locations. When tested using the model-predicted at-stack meteorological values, and NPRI stack emissions data, the Briggs parameterization and the layered approach resulted in very different plume rise behaviour. Predicted surface $SO_2$ concentration performance was substantially improved across all metrics when the layered approach was used, and aircraft $SO_2$ comparisons improved for all metrics aside from bias. For the predicted plume heights, the slope of the model observation line was -0.16 for the Briggs parameterization, and 0.97 for the layered approach, with the former under-predicting, and the latter over-predicting the aircraft-observation-estimated plume height. The reader is directed to Akingunola *et al.* (2018) for a discussion of these issues, which suggests that accuracy of estimates of the driving meteorological parameters at the stack locations has a controlling influence on the performance of the layered approach, and with the layered approach recommended for future development. "

Non-public comments to the Author:

Some minor technical issues I found. A careful proof-read of the final manuscript would be helpful as I have not gone through this thoroughly.

All the minor corrections listed below have been incorporated and the issue has been proof-read again.

L144: "The atmosphere is considered .." not "the plume is .."

L196: "found" not "hound"

Table 1 caption: "the flight" not "flight"

Line 489: Consider saying "Non-stationarity" instead of "Stationarity" or say: "The assumption of stationarity..."

Line 513: the variable 's' is used for two different things. This is a bit confusing.

Fig 4 caption: "data" not "date"

L723: Seems more appropriate to say "estimated" not "calculated"

Great work!